# The effect of the zero-field splitting in light-induced pulsed dipolar EPR spectroscopy

Andreas Scherer, Berk Yildirim, Malte Drescher

Department of Chemistry and Konstanz Research School Chemical Biology, University of Konstanz, 78457 Konstanz, Germany

*Correspondence to*: Malte Drescher (malte.drescher@uni-konstanz.de)

**Abstract.** Laser-induced magnetic dipole (LaserIMD) spectroscopy and light-induced double electron-electron resonance (LiDEER) spectroscopy are important techniques in the emerging field of light-induced pulsed dipolar EPR spectroscopy (light-induced PDS). These techniques use the photoexcitation of a chromophore to the triplet state and measure its dipolar coupling to a neighboring electron spin, which allows the determination of distance restraints. LaserIMD and LiDEER were so far analyzed with software tools that were developed for a pair of two $S = 1/2$ spins and neglect the zero-field splitting (ZFS) of the excited triplet. Here, we explore the limits of this assumption and show that the ZFS can have a significant effect on the shape of the dipolar trace. For a detailed understanding of the effect of the ZFS, a theoretical description for LaserIMD and LiDEER is derived, taking into account the non-secular terms of the ZFS. Simulations based on this model show that the effect of the ZFS is not so pronounced in LiDEER for experimentally relevant conditions. However, the ZFS leads to an additional decay in the dipolar trace in LaserIMD. This decay is not so pronounced in Q-band but can be quite noticeable for lower magnetic field strengths in X-band. Experimentally recorded LiDEER and LaserIMD data confirm these findings. It is shown that ignoring the ZFS in the data analysis of LaserIMD traces can lead to errors in the obtained modulation depths and background decays. In X-band, it is additionally possible that the obtained distance distribution is plagued by long distance artifacts.

## 1 Introduction

Pulsed dipolar EPR spectroscopy (PDS) has become an important tool for nanoscale distance determination in soft matter. Its applications include the structural determination of biomacromolecules like proteins (Yee et al., 2015; Yang et al., 2020; Giannoulis et al., 2020; Weickert et al., 2020; Robotta et al., 2014; Ritsch et al., 2022), DNA (Wojciechowski et al., 2015; Takeda et al., 2004; Marko et al., 2011) and RNA (Collauto et al., 2020), but also synthetic polymers (Jeschke et al., 2010) as well as nanoparticles (Hintze et al., 2015; Bücker et al., 2019). PDS measures the dipolar coupling between two spin centers within the molecule under investigation. Oftentimes, the spin centers need to be introduced as spin labels via site-directed labeling, with nitroxide spin probes as the most common example (Hubbell et al., 2013; Roser et al., 2016; García-Rubio, 2020). The most common PDS technique is double electron-electron resonance (DEER, also called PELDOR) spectroscopy (Milov et al., 1981, 1984; Jeschke, 2012). Here, one of the spin labels is excited by microwave pulses at an observer frequency

to generate a refocused echo. The excitation of the other spin label by a pump pulse at a second frequency leads to an oscillation of the refocused echo, when the pump pulse is shifted in the time domain. The frequency of this oscillation depends on the inverse cubic distance between the spin labels $r^{-3}$ and thus provides distance information for the molecule under investigation (Jeschke, 2012).

The recent years have seen an advent of a new type of spin label, which are in an EPR-silent singlet ground state, but can be converted transiently to a triplet state by photoexcitation and subsequent inter-system crossing (Di Valentin et al., 2014; Bertran et al., 2022a). In contrast to spin labels with a spin of $S = 1/2$ like nitroxides, these transient triplet labels are subject to an additional zero-field splitting (ZFS). It is described by the ZFS parameters $D$ and $E$. By now, several transient triplet labels with different ZFS strengths have been used. Examples are triphenylporphyrin (TPP) ($D = 1159\,\text{MHz}$, $E = -238\,\text{MHz}$ ) (Di

Valentin et al., 2014), fullerenes ($D = 342\,\text{MHz}$, $E = -2\,\text{MHz}$ ) (Wasielewski et al., 1991; Krumkacheva et al., 2019; Timofeev et al., 2022), Rose Bengal ($D = 3671\,\text{MHz}$, $E = -319\,\text{MHz}$ ), Eosin Y ($D = 2054\,\text{MHz}$, $E = -585\,\text{MHz}$ ), Atto Thio12 ($D = 1638\,\text{MHz}$, $E = -375\,\text{MHz}$ ) (Serrer et al., 2019; Williams et al., 2020) and Erythrosin B ($D = 3486\,\text{MHz}$, $E = -328\,\text{MHz}$ ) (Bertran et al., 2022b). The most common PDS techniques for transient triplet labels are light-induced DEER (LiDEER) and laser-induced magnetic dipole (LaserIMD) spectroscopy (Di Valentin et al., 2014; Hintze et al., 2016). They

both allow the determination of distances between one permanent spin label and one transient triplet label. LiDEER is a modification of DEER with an additional laser flash preceding the microwave pulses (see Figure 1a). The permanent spin is excited by the pump pulse, because it typically has an EPR spectrum that is narrower than the one of the transient triplet label, which gives higher modulation depths. The transient triplet label is observed, because despite its broader EPR spectrum it is still possible to generate strong echoes, because the photoexcitation of the transient triplet label typically leads to a high spin

polarization (Di Valentin et al., 2014). In LaserIMD, on the other hand, the permanent spin label is observed. During the evolution of the observer spin, the transient triplet label is excited by a laser flash (see Figure 1b). The induced transition from the singlet to the triplet state has the equivalent effect as the microwave pump pulse in DEER and results in an oscillation of the echo of the observer spin. An advantage of LaserIMD is that, in contrast to DEER, the bandwidth of the laser excitation is neither limited by the width of the EPR spectrum of the pump spin nor the resonator bandwidth. This gives virtually infinite

excitation bandwidths and promises high modulation depths also in cases where the microwave excitation bandwidth is smaller than the EPR spectra of the invoked spins (Scherer et al., 2022).

## a) LiDEER

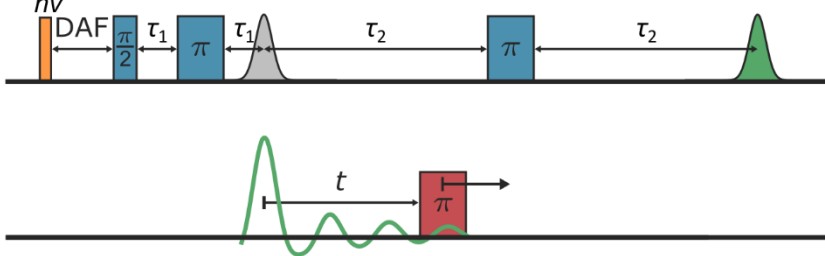

## b) LaserIMD

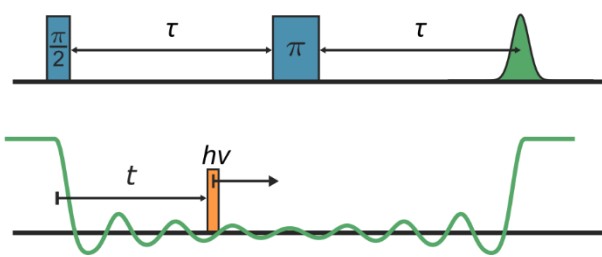

**Figure 1:** The pulse sequences of a) LiDEER and b) LaserIMD. The observed green echoes are modulated when the pump pulse (LiDEER) or laser flash (LaserIMD) is shifted in the time domain.

In previous works, LaserIMD and LiDEER data were analyzed under the assumption that the ZFS of the transient triplet label can be ignored (Di Valentin et al., 2014; Hintze et al., 2016; Bieber et al., 2018; Dal Farra et al., 2019a; Krumkacheva et al., 2019). Under this assumption the dipolar traces of LaserIMD and LiDEER have the same shape as those of DEER on a label pair with two $S = 1/2$ spins. However, as is shown below, this assumption is only correct if all spin-spin interactions are much smaller than the Zeeman-interaction with the external magnetic field. Then, all non-secular terms in the Hamiltonian can be dropped (Manukovsky et al., 2017). The excited triplet state of transient triplet labels with a total spin of $S = 1$, however, can be subject to a strong ZFS, reaching values over 1 GHz in many cases (Di Valentin et al., 2014; Williams et al., 2020). For other high-spin labels like Gd$^{III}$ or high-spin Fe$^{III}$, it is already known that the ZFS can have an effect on the recorded dipolar trace and that it has to be included in the data-analysis routine, if artifacts in the distance shall be avoided (Maryasov et al., 2006; Dalaloyan et al., 2015; Abdullin et al., 2019).

Here, we set out to investigate the effect of the ZFS in light-induced PDS. Therefore, we are going to derive a theoretical description for light-induced PDS taking the $S = 1$ spin state and ZFS of the triplet state into account. Section 3 will report about the materials and methods used. In section 4, the theoretical model will be used for numerical simulations of LaserIMD and time-domain simulations were performed for LiDEER. It will be shown that in both methods, but particularly in LaserIMD, the effect of the ZFS can result in significant differences in the dipolar traces compared to the $S = 1/2$ case where the ZFS is

ignored. In section 5, experimental LaserIMD and LiDEER traces are shown and the influence of the ZFS is discussed by comparing the model with the experimental data.

## 2 Theoretical derivation

### 2.1 DEER

For the analysis of DEER data, one typically uses the assumption that both spins are of $S = 1/2$ nature and the system is in high-field and weak-coupling limit so that all pseudo- and non-secular parts of the spin Hamiltonian can be dropped (Jeschke et al., 2006; Worswick et al., 2018; Fábregas Ibáñez et al., 2020). In this case, there are two coherence transfer pathways that contribute to the DEER signal; one where the pump spin is flipped from the state with $m_S = +1/2$ to $m_S = -1/2$, and the one where it is flipped from $m_S = -1/2$ to $m_S = +1/2$. The frequency of the dipolar oscillation of the refocused echo for the

two coherence transfer pathways is:

$$\omega_{\mathrm{DEER}, +\frac{1}{2} \to -\frac{1}{2}} = \left(3\cos(\beta_{\mathrm{dip}})^2 - 1\right)\omega_{\mathrm{dip}}, \tag{1}$$

$$\omega_{\mathrm{DEER}, -\frac{1}{2} \to +\frac{1}{2}} = -\left(3\cos(\beta_{\mathrm{dip}})^2 - 1\right)\omega_{\mathrm{dip}}. \tag{2}$$

Here, $\beta_{\mathrm{dip}}$ is the angle between the dipolar coupling vector and the external magnetic field and $\omega_{\mathrm{dip}}$ is the dipolar coupling in radial frequency units. It depends on the distance $r$ between the two labels:

$$\omega_{\mathrm{dip}} = \frac{\mu_{\mathrm{B}}^2 g_1 g_2}{\hbar} \frac{1}{r^3}, \tag{3}$$

With the Bohr-magneton $\mu_{\mathrm{B}}$, the reduced Plank constant $\hbar$, and the $g$-values $g_1$ and $g_2$ of the two spin labels. In experiments one typically measures powder samples, thus molecules with all orientations with respect to the external field contribute to the

signal, and the weighted integral over all angles $\beta_{\mathrm{dip}}$ must be taken (Pake, 1948; Milov et al., 1998). In the high-temperature limit, which is often fulfilled in experiments, the population of the spin states with $m_S = +1/2$ to $m_S = -1/2$ is virtually identical and therefore both coherence transfer pathways contribute equally to the signal (Marko et al., 2013). In this case the integral over all orientations is:

$$S_{\mathrm{DEER}}(t, r) = \int_0^{\pi/2} \mathrm{d}\beta_{\mathrm{dip}} \sin(\beta_{\mathrm{dip}}) \cos\left(t\left(3\cos(\beta_{\mathrm{dip}})^2 - 1\right)\omega_{\mathrm{dip}}(r)\right) \tag{4}$$

Here, $t$ is the time at which the pump pulse flips the pump spins. Due to a limited excitation bandwidth and pulse imperfections

not all spins can be excited by the pump pulse, therefore a part of the signal is not modulated:

$$F_{\mathrm{DEER}}(t, r) = \lambda S_{\mathrm{DEER}}(t, r) + (1 - \lambda), \tag{5}$$

where the modulation depth $\lambda$ depends on the fraction of excited pump spins. The experimental signal is the product of this intramolecular contribution $F_{\mathrm{DEER}}(t, r)$ and a contribution from the intermolecular dipolar interactions $B(t)$, that is typically

termed background. Finally, the contributions from all distances need to be included by integrating over the distance distribution $P(r)$:

$$V_{\text{DEER}}(t) = \int \mathrm{d}r K_{\text{DEER}}(t,r)P(r) = \int \mathrm{d}r B(t)F_{\text{DEER}}(t,r)P(r). \tag{6}$$

The kernel $K_{\text{DEER}}(t,r)$ describes the relation between the distance distribution and the measured dipolar trace in DEER. In a sample with a homogenous distribution of spins, the background function can be obtained by integrating over all dipolar interactions within the sample, which results in (Hu and Hartmann, 1974):

$$B(t) = \exp(-k|t|). \tag{7}$$

The decay constant $k$ is proportional to the spin concentration and modulation depth (Hu and Hartmann, 1974). By inverting Eq. (6), it is possible to extract the distance distribution $P(r)$ from the experimentally recorded signal $V_{\text{DEER}}(t)$. Because this is an ill-posed problem, this is typically done by advanced techniques like Tikhonov regularization (Bowman et al., 2004; Jeschke et al., 2004) or neural networks (Worswick et al., 2018; Keeley et al., 2022).

## 2.2 LaserIMD

In LaserIMD, the spin system consists of a permanent spin label, which serves as an observer spin, and a transient triplet label, which is excited by a laser flash. In many cases, the permanent spin label is or can be assumed to be a doublet with $S_{\text{D}} = 1/2$. Before the photoexcitation, the transient label is still in its singlet state and does therefore neither interact with the external field $B$, nor with the doublet $S_{\text{D}}$. The Hamiltonian thus only contains the Zeeman interaction of $S_{\text{D}}$:

$$\hat{H}_{\text{dark}} = 2\pi\nu_{\text{D}}\hat{S}_{\text{D},z}, \tag{8}$$

with the Zeeman frequency $\nu_{\text{D}} = \frac{g_{\text{D}}\mu_{\text{B}}}{2\pi\hbar}B$, $g_{\text{D}}$ the $g$-values of $S_{\text{D}}$, which is assumed to be isotropic. The Hamiltonian is written in units of radial frequencies. This Hamiltonian has two eigenvalues:

$$E_{+\frac{1}{2}, \text{dark}} = \frac{2\pi\nu_{\text{D}}}{2}, \tag{9}$$

$$E_{-\frac{1}{2}, \text{dark}} = -\frac{2\pi\nu_{\text{D}}}{2}. \tag{10}$$

When the laser flash excites the transient triplet label to the triplet state $S_{\text{T}} = 1$, the Zeeman interaction of $S_{\text{T}}$, the ZFS between the two unpaired electrons that form the triplet $S_{\text{T}}$ and the dipolar coupling between $S_{\text{D}}$ and $S_{\text{T}}$ has to be included in the Hamiltonian:

$$\hat{H} = 2\pi\nu_{\text{D}}\hat{S}_{\text{D},z} + 2\pi\nu_{\text{T}}\hat{S}_{\text{T},z} + \boldsymbol{S}_{\text{T}} \cdot \mathbf{D} \cdot \boldsymbol{S}_{\text{T}} + \boldsymbol{S}_{\text{T}} \cdot \mathbf{T} \cdot \boldsymbol{S}_{\text{D}}. \tag{11}$$

Here, $\nu_{\text{T}} = \frac{g_{\text{T}}\mu_{\text{B}}B}{2\pi\hbar}$ is the Zeeman frequency of the spin $S_{\text{T}}$ with its isotropic $g$-value $g_{\text{T}}$. $\bar{S}_{\text{D}}$ and $\bar{S}_{\text{T}}$ represent the vectors of the Cartesian spin operators $\boldsymbol{S}_{\text{D}} = (\hat{S}_{\text{D},x}, \hat{S}_{\text{D},y}, \hat{S}_{\text{D},z})^{\text{T}}$ and $\boldsymbol{S}_{\text{T}} = (\hat{S}_{\text{T},x}, \hat{S}_{\text{T},y}, \hat{S}_{\text{T},z})^{\text{T}}$. The ZFS tensor $\mathbf{D}$ is described by the ZFS values $D = \frac{3}{2}D_z$ and $E = \frac{D_x - D_y}{2}$, where $D_x, D_y$ and $D_z$ are the eigenvalues of the ZFS tensor (Telser, 2017). Its orientation is described by the three Euler angles $\alpha_T, \beta_T$ and $\gamma_T$ that connect the laboratory frame with the molecular frame of the transient triplet label.

In the point-dipole approximation, the dipolar coupling tensor **T** is axial with the eigenvalues $T_x = T_y = -\omega_{\mathrm{dip}}$ and $T_z = 2\omega_{\mathrm{dip}}$ (Schweiger and Jeschke, 2001). Its orientation towards the external magnetic field is described by the angle $\beta_{\mathrm{dip}}$.

In the high-field and weak-coupling limit all non- and pseudo-secular terms can be dropped from the Hamiltonian. The remaining secular Hamiltonian (see Eq. (S2) in S1) is already diagonal in the high-field basis with the energy levels $E^{\mathrm{sec}}_{m_{\mathrm{D}},m_{\mathrm{T}}}$, where $m_{\mathrm{D}}$ and $m_{\mathrm{T}}$ are the magnetic quantum numbers of the doublet $S_{\mathrm{D}}$ and the triplet $S_{\mathrm{T}}$. The exact expressions for the energies $E^{\mathrm{sec}}_{m_{\mathrm{D}},m_{\mathrm{T}}}$ can be found in Eq. (S4) -(S9) in S1. In LaserIMD, the initial $\frac{\pi}{2}$-pulse generates a coherence of the observer spin $S_{\mathrm{D}}$. Before the laser excitation, the coherence evolves with a frequency of $E_{+\frac{1}{2},\,\mathrm{dark}} - E_{-\frac{1}{2},\,\mathrm{dark}} = 2\pi\nu_{\mathrm{D}}$, it is not influenced by the dipolar coupling because the transient triplet label is still in a singlet state with $S_{\mathrm{T}} = 0$ and $m_{\mathrm{T}} = 0$. The excitation of the transient triplet label leads to three different coherence transfer pathways, depending to which manifold $m_{\mathrm{T}} = 1, 0$ or $-1$ of the triplet the transient label is excited to. Depending on the triplet state $m_{\mathrm{T}}$, the coherence will then continue to evolve with $E^{\mathrm{sec}}_{+\frac{1}{2},\,m_{\mathrm{T}}} - E^{\mathrm{sec}}_{-\frac{1}{2},\,m_{\mathrm{T}}}$. The refocusing $\pi$-pulse generates an echo at the time $2\tau$. Due to the different frequencies before and after the excitation at a variable time $t$, the coherences are not completely refocused but depending on the time of the laser flash they will have gained a phase $\phi = \omega^{\mathrm{sec}}_{m_{\mathrm{T}}}t$, that depends on the LaserIMD frequency $\omega^{\mathrm{sec}}_{m_{\mathrm{T}}}$ of the corresponding triplet manifold $m_{\mathrm{T}}$. When only the secular terms are considered in the Hamiltonian, the LaserIMD frequencies $\omega^{\mathrm{sec}}_{m_{\mathrm{T}}}$ do not depend on the ZFS, because its secular terms cancel each other out and the same expression as by (Hintze et al., 2016) are obtained:

$$\omega^{\mathrm{sec}}_{+1} = \left(E^{\mathrm{sec}}_{+\frac{1}{2},+1} - E^{\mathrm{sec}}_{-\frac{1}{2},+1}\right) - \left(E_{+\frac{1}{2},\,\mathrm{dark}} - E_{-\frac{1}{2},\,\mathrm{dark}}\right) = \left(3\cos(\beta_{\mathrm{dip}})^2 - 1\right)\omega_{\mathrm{dip}}, \tag{12}$$

$$\omega^{\mathrm{sec}}_{0} = \left(E^{\mathrm{sec}}_{+\frac{1}{2},\,0} - E^{\mathrm{sec}}_{-\frac{1}{2},\,0}\right) - \left(E_{+\frac{1}{2},\,\mathrm{dark}} - E_{-\frac{1}{2},\,\mathrm{dark}}\right) = 0, \tag{13}$$

$$\omega^{\mathrm{sec}}_{-1} = \left(E^{\mathrm{sec}}_{+\frac{1}{2},-1} - E^{\mathrm{sec}}_{-\frac{1}{2},-1}\right) - \left(E_{+\frac{1}{2},\,\mathrm{dark}} - E_{-\frac{1}{2},\,\mathrm{dark}}\right) = -\left(3\cos(\beta_{\mathrm{dip}})^2 - 1\right)\omega_{\mathrm{dip}}. \tag{14}$$

When the transient triplet label is excited to $m_{\mathrm{T}} = 1$ or $m_{\mathrm{T}} = -1$, the LaserIMD frequencies in secular-approximation from Eq. (12) and (14) are identical to the DEER frequencies in Eq. (1) and (2). Here, the laser flash leads to a change in the magnetic quantum number of $\Delta m_{\mathrm{T}} = \pm 1$, which is equivalent to the effect of the microwave pump pulse in DEER. In the case when the transient triplet label is excited to the state $m_{\mathrm{T}} = 0$, however, the secular approximation predicts that the echo is not oscillating, because -loosely spoken- there is no change in the magnetic spin quantum number of the transient triplet label, which means that the dipolar coupling is not changed. Like it is the case in DEER, the measured signal is the average over all orientations of the spin system. Whereas in DEER it is only necessary to consider the orientation of the dipolar vector, in LaserIMD the orientation of the transient triplet label must also be taken into account, therefore it is necessary to also integrate over the three corresponding Euler angles $\alpha_{\mathrm{T}}$, $\beta_{\mathrm{T}}$ and $\gamma_{\mathrm{T}}$ (Bak and Nielsen, 1997). In absence of orientation selection, the orientation of the dipolar vector and the transient triplet label are not correlated and the integration over the corresponding Euler angles can be done independently. This is often realized in practical applications where flexible linkers are used to attach labels to the studied molecule. As the triplet state of the transient label is reached by intersystem-crossing, the population of the three high-field

triplet states $m_T = +1, 0, -1$ depends on the orientation of the transient label with respect to the external magnetic field and the populations $P_x$, $P_y$ and $P_z$ of the zero-field eigenstates (Rose, 1995). The contribution of the three coherence transfer pathways must be weighted by population of these high-field states; this gives (still in secular approximation) the three expressions:

$$S_{+1}^{\text{sec}}(t,r) = \frac{1}{8\pi^2} \int_0^{2\pi} d\alpha_T \int_0^{\pi} d\beta_T \sin(\beta_T) \int_0^{2\pi} d\gamma_T \left( \frac{P_z}{2} \sin^2(\beta_T) + \frac{P_x}{2} (\cos^2(\beta_T) + \sin^2(\beta_T) \sin^2(\gamma_T)) \right.$$

$$\left. + \frac{P_y}{2} (\cos^2(\beta_T) + \sin^2(\beta_T) \cos^2(\gamma_T)) \right) \int_0^{\frac{\pi}{2}} d\beta_{\text{dip}} \sin(\beta_{\text{dip}}) \exp(-i\omega_{+1}^{\text{sec}}(\beta_{\text{dip}})t),$$

(15)

$$S_0^{\text{sec}}(t,r) = \frac{1}{8\pi^2} \int_0^{2\pi} d\alpha_T \int_0^{\pi} d\beta_T \sin(\beta_T) \int_0^{2\pi} d\gamma_T (P_z \cos^2(\beta_T) + P_x \sin^2(\beta_T) \cos^2(\gamma_T)$$

$$+ P_y \sin^2(\beta_T) \sin^2(\gamma_T)) \int_0^{\pi/2} d\beta_{\text{dip}} \sin(\beta_{\text{dip}}) \exp(-i\omega_0^{\text{sec}}(\beta_{\text{dip}})t),$$

(16)

$$S_{-1}^{\text{sec}}(t,r) = \frac{1}{8\pi^2} \int_0^{2\pi} d\alpha_T \int_0^{\pi} d\beta_T \sin(\beta_T) \int_0^{2\pi} d\gamma_T \left( \frac{P_z}{2} \sin^2(\beta_T) + \frac{P_x}{2} (\cos^2(\beta_T) + \sin^2(\beta_T) \sin^2(\gamma_T)) \right.$$

$$\left. + \frac{P_y}{2} (\cos^2(\beta_T) + \sin^2(\beta_T) \cos^2(\gamma_T)) \right) \int_0^{\frac{\pi}{2}} d\beta_{\text{dip}} \sin(\beta_{\text{dip}}) \exp(-i\omega_{-1}^{\text{sec}}(\beta_{\text{dip}})t).$$

(17)

5 Performing the integration over the orientations of the transient label $\alpha_T$, $\beta_T$ and $\gamma_T$ and taking the sum gives (Williams et al., 2020):

$$S_{\text{LaserIMD}}^{\text{sec}}(t,r) = S_{+1}^{\text{sec}}(t,r) + S_0^{\text{sec}}(t,r) + S_{-1}^{\text{sec}}(t,r) = \frac{2}{3} \int_0^{\pi/2} \cos\big(\omega_{\text{dip}}(3\cos^2(\beta_{\text{dip}}) - 1)t\big) \sin(\beta_{\text{dip}}) \, d\beta_{\text{dip}} + \frac{1}{3}$$

$$= \frac{2}{3} S_{\text{DEER}}(t,r) + \frac{1}{3}.$$

(18)

In secular-approximation, the first term of the LaserIMD signal is equivalent to the trace $S_{\text{DEER}}(t)$ (Edwards and Stoll, 2018). The second term is an additional non-modulated contribution. For the final expression for the kernel $K_{\text{LaserIMD}}^{\text{sec}}(t,r)$, the quantum yield of the triplet state is considered by an additional factor $\gamma$ and the intermolecular interaction to other spins in the

10 sample has to be considered as background $B(t)$:

$$K_{\text{LaserIMD}}^{\text{sec}}(t,r) = B(t)(\gamma S_{\text{LaserIMD}}^{\text{sec}}(t,r) + 1 - \gamma).$$

(19)

This can be rewritten as:

$$K_{\text{LaserIMD}}^{\text{sec}}(t,r) = B(t)(\lambda S_{\text{DEER}}(t,r) + 1 - \lambda),$$

(20)

with the modulation depth $\lambda = 2/3\gamma$. The only difference between LaserIMD in the secular approximation and DEER is that in LaserIMD, even for a triplet yield of $\gamma = 100\%$, there is coherence transfer pathway with $\Delta m_S = 0$ that does not result in a dipolar oscillation, which limits the maximum achievable modulation depth to $66.\overline{6}\%$. The calculations so far show that if

the secular approximation can be employed, the ZFS has no effect on the LaserIMD trace and it is possible to analyze experimentally recorded LaserIMD data with the same kernel that can be used for DEER.

Even though in the secular approximation the ZFS has no effect in LaserIMD, it cannot be taken for granted that the non-secular terms can be ignored because the ZFS of some transient triplet labels can be quite large (Williams et al., 2020). Here, we additionally consider the terms $\hat{S}_{T,z}\hat{S}_{T,+} + \hat{S}_{T,+}\hat{S}_{T,z}$ and $\hat{S}_{T,-}\hat{S}_{T,z} + \hat{S}_{T,-}\hat{S}_{T,z}$ from the ZFS interaction and the terms $\hat{S}_{D,z}\hat{S}_{T,+}$ and $\hat{S}_{D,z}\hat{S}_{T,-}$ from the dipolar coupling. They connect the adjacent triplet states $|+1\rangle$ and $|0\rangle$ and $|0\rangle$ and $|-1\rangle$ of the triplet manifold and shift their energy in second order (Hagston and Holmes, 1980). This is illustrated in Figure 2. The details of this calculation are described in S1. For this calculation, the remaining ZFS terms $\hat{S}_{T,+}^2$ and $\hat{S}_{T,-}^2$ were ignored. They connect the triplet states $|+1\rangle$ and $|-1\rangle$, which have a larger energy difference than adjacent states. Therefore, the second order energy shift of $\hat{S}_{T,+}^2$ and $\hat{S}_{T,-}^2$ is weaker than those of the considered terms. The terms $\hat{S}_{D,+}\hat{S}_{T,+}$, $\hat{S}_{D,-}\hat{S}_{T,+}$, $\hat{S}_{D,+}\hat{S}_{T,-}$, $\hat{S}_{D,-}\hat{S}_{T,-}$, $\hat{S}_{D,+}\hat{S}_{T,z}$ and $\hat{S}_{D,-}\hat{S}_{T,z}$ of the dipolar coupling were also ignored. They connect spin states of different manifolds of the doublet spin and the corresponding energies cannot be significantly shifted by the comparably weak dipolar coupling. It is shown in S2 that at magnetic field strengths that are relevant for experimental conditions the included non-secular terms from Eq. (S3) are sufficient and no further distortions are to be expected by the left-out ones.

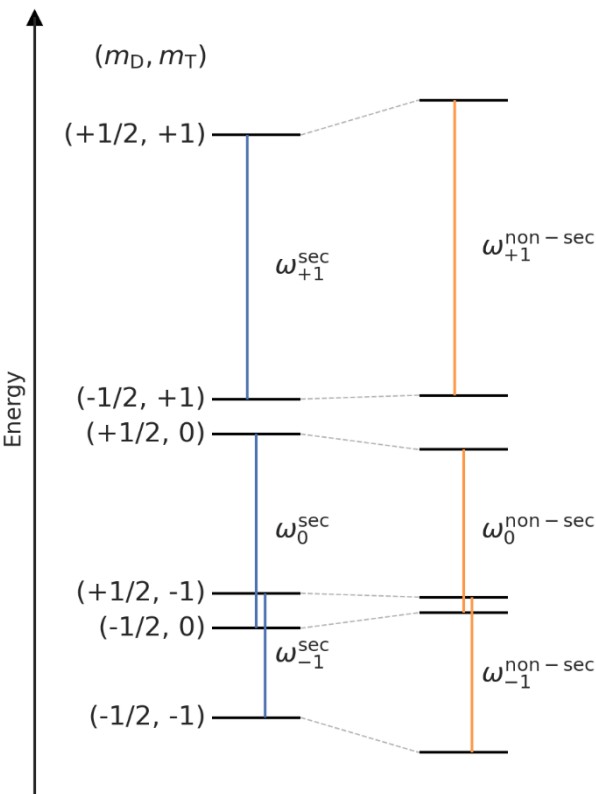

**Figure 2:** Energy level diagram (not to scale) after the transient triplet label has been excited to the triplet state demonstrating the shift that is induced by the non-secular terms of the ZFS and dipolar coupling from Eq. (S3). The energy levels in secular approximation are shown on the left and the levels with the non-secular terms are shown on the right. The vertical lines in blue (secular approximation) and orange (non-secular terms included) indicate the coherences of the permanent spin label that are excited during the LaserIMD pulse sequence. They are marked with the corresponding transition frequencies.

The shift of the energy levels also leads to a shift in the LaserIMD frequencies (see S1):

$$\omega_{+1}^{\text{non-sec}} = \left(E_{+\frac{1}{2}, +1}^{\text{non-sec}} - E_{-\frac{1}{2}, +1}^{\text{non-sec}}\right) - \left(E_{+\frac{1}{2}, \text{dark}} - E_{-\frac{1}{2}, \text{dark}}\right) = \left(\left(3\cos(\beta_{\text{dip}})^2 - 1\right) + \delta_{\text{ZFS}} \sin(2\beta_{\text{dip}})\right) \omega_{\text{dip}}, \tag{21}$$

$$\omega_{0}^{\text{non-sec}} = \left(E_{+\frac{1}{2}, 0}^{\text{non-sec}} - E_{-\frac{1}{2}, 0}^{\text{non-sec}}\right) - \left(E_{+\frac{1}{2}, \text{dark}} - E_{-\frac{1}{2}, \text{dark}}\right) = -2\delta_{\text{ZFS}} \sin(2\beta_{\text{dip}}) \omega_{\text{dip}}, \tag{22}$$

$$\omega_{-1}^{\text{non-sec}} = \left(E_{+\frac{1}{2}, -1}^{\text{non-sec}} - E_{-\frac{1}{2}, -1}^{\text{non-sec}}\right) - \left(E_{+\frac{1}{2}, \text{dark}} - E_{-\frac{1}{2}, \text{dark}}\right) = \left(-\left(3\cos(\beta_{\text{dip}})^2 - 1\right) + \delta_{\text{ZFS}} \sin(2\beta_{\text{dip}})\right) \omega_{\text{dip}}, \tag{23}$$

with

$$\delta_{\text{ZFS}} = \frac{3\sin(2\beta_{\text{T}})\cos(\alpha_{\text{T}})\, D - 6\sin(\beta_{\text{T}})(\cos(\beta_{\text{T}})\cos(2\gamma_{\text{T}})\cos(\alpha_{\text{T}}) - \sin(2\gamma_{\text{T}})\sin(\alpha_{\text{T}}))E}{8\pi\nu_{\text{T}}}. \tag{24}$$

As can be seen from Eq. (21) -(23), the frequencies $\omega_{+1}^{\text{non-sec}}$ and $\omega_{-1}^{\text{non-sec}}$ are the sum of the unperturbed frequencies $\omega_{+1}^{\text{sec}}$ and $\omega_{-1}^{\text{sec}}$ and a frequency shift $\delta_{\text{ZFS}}\sin(2\beta_{\text{dip}})\omega_{\text{dip}}$, which contains the effect of the ZFS. Most notably, the coherence transfer pathway with $\Delta m_{\text{T}} = 0$ does not lead to a vanishing LaserIMD frequency as it was the case in the secular approximation. Instead, we find that $\omega_0^{\text{non-sec}}$ equals twice the negative of the frequency shift that is experienced by the other two coherence transfer pathways. The frequency shift scales with $\delta_{\text{ZFS}}$ that depends on the ZFS values $D$ and $E$, the Zeeman frequency of the transient triplet label $\omega_{\text{T}}$ as well as the orientation of the transient triplet label, described by $\alpha_{\text{T}}$, $\beta_{\text{T}}$ and $\gamma_{\text{T}}$. At a higher ZFS and a smaller magnetic field, the shift of the LaserIMD frequencies will be larger, so that larger disturbances in the LaserIMD trace can be expected in these cases.

The powder average is more complex when the non-secular terms are included, because the LaserIMD frequencies now also depend on the orientation of the transient triplet label. Still assuming no orientation selection, this gives the following integrals:

$$S_{+1}^{\text{non-sec}}(t,r) = \frac{1}{8\pi^2}\int_0^{2\pi}\mathrm{d}\alpha_{\text{T}}\int_0^{\pi}\mathrm{d}\beta_{\text{t}}\sin(\beta_{\text{T}})\int_0^{2\pi}\mathrm{d}\gamma_{\text{T}}\left(\frac{P_z}{2}\sin^2(\beta_{\text{T}}) + \frac{P_x}{2}(\cos^2(\beta_{\text{T}}) + \sin^2(\beta_{\text{T}})\sin^2(\gamma_{\text{T}}))\right.$$
$$\left. + \frac{P_y}{2}(\cos^2(\beta_{\text{T}}) + \sin^2(\beta_{\text{T}})\cos^2(\gamma_{\text{T}}))\right)\int_0^{\frac{\pi}{2}}\mathrm{d}\beta_{\text{dip}}\sin(\beta_{\text{dip}})\exp(-i\omega_{+1}^{\text{non-sec}}(\alpha_{\text{T}},\beta_{\text{T}},\gamma_{\text{T}},\beta_{\text{dip}})t), \tag{25}$$

$$S_0^{\text{non-sec}}(t,r) = \frac{1}{8\pi^2}\int_0^{2\pi}\mathrm{d}\alpha_{\text{T}}\int_0^{\pi}\mathrm{d}\beta_{\text{T}}\sin(\beta_{\text{T}})\int_0^{2\pi}\mathrm{d}\gamma_{\text{T}}\left(P_z\cos^2(\beta_{\text{T}}) + P_x\sin^2(\beta_{\text{T}})\cos^2(\gamma_{\text{T}})\right.$$
$$\left. + P_y\sin^2(\beta_{\text{T}})\sin^2(\gamma_{\text{T}})\right)\int_0^{\pi/2}\mathrm{d}\beta_{\text{dip}}\sin(\beta_{\text{dip}})\exp(-i\omega_0^{\text{non-sec}}(\alpha_{\text{T}},\beta_{\text{T}},\gamma_{\text{T}},\beta_{\text{dip}})t), \tag{26}$$

$$S_{-1}^{\text{non-sec}}(t,r) = \frac{1}{8\pi^2}\int_0^{2\pi}\mathrm{d}\alpha_{\text{T}}\int_0^{\pi}\mathrm{d}\beta_{\text{T}}\sin(\beta_{\text{t}})\int_0^{2\pi}\mathrm{d}\gamma_{\text{T}}\left(\frac{P_z}{2}\sin^2(\beta_{\text{T}}) + \frac{P_x}{2}(\cos^2(\beta_{\text{T}}) + \sin^2(\beta_{\text{T}})\sin^2(\gamma_{\text{T}}))\right.$$
$$\left. + \frac{P_y}{2}(\cos^2(\beta_{\text{T}}) + \sin^2(\beta_{\text{T}})\cos^2(\gamma_{\text{T}}))\right)\int_0^{\frac{\pi}{2}}\mathrm{d}\beta_{\text{dip}}\sin(\beta_{\text{dip}})\exp(-i\omega_{-1}^{\text{non-sec}}(\alpha_{\text{T}},\beta_{\text{T}},\gamma_{\text{T}},\beta_{\text{dip}})t). \tag{27}$$

The sum over these terms gives the final intramolecular contribution in LaserIMD.

$$S_{\text{LaserIMD}}^{\text{non-sec}}(t) = S_{+1}^{\text{non-sec}}(t) + S_0^{\text{non-sec}}(t) + S_{-1}^{\text{non-sec}}(t). \tag{28}$$

By including incomplete excitation and the intermolecular dipolar interactions, one arrives at the final model

$$K_{\text{LaserIMD}}^{\text{non-sec}}(t,r) = B(t)(\lambda S_{\text{LaserIMD}}^{\text{non-sec}}(t,r) + 1 - \lambda). \tag{29}$$

Unlike it was the case for the secular approximation, the integrals are difficult to solve analytically and further insight in this expression will be gained by numerical integrations in the next sections. However, it can already be seen without further calculations that with the non-secular terms the ZFS has an influence in LaserIMD and that the resulting kernel no longer corresponds to the kernel $K_{\text{DEER}}(t,r)$ of the $S = 1/2$ case.

## 2.3 LiDEER

In LiDEER, the transient triplet label is observed and the permanent spin label is pumped. For simplicity, we will derive the expressions within the secular approximation first and afterwards turn to the case that includes the non-secular terms. Due to the limited excitation bandwidth of the observer pulse, either the transition between the states with $m_T = 1$ and $m_T = 0$ or the states with $m_T = 0$ and $m_T = -1$ of the transient triplet label is excited. If the transition between the states $m_T = 1$ and $m_T = 0$ is excited, the excited coherence of the triplet spin will either evolve with the frequency $\omega^{\text{sec}}_{+\frac{1}{2}, 1\leftrightarrow 0} = E^{\text{sec}}_{+\frac{1}{2}, +1} - E^{\text{sec}}_{+\frac{1}{2}, 0}$ or $\omega^{\text{sec}}_{-\frac{1}{2}, 1\leftrightarrow 0} = E^{\text{sec}}_{-\frac{1}{2}, +1} - E^{\text{sec}}_{-\frac{1}{2}, 0}$, depending on whether the permanent spin label is in the state with $m_D = 1/2$ or $m_D = -1/2$. Pumping the permanent spin label at the time $t$ will result in a transition from $m_D = +\frac{1}{2}$ to $m_D = -\frac{1}{2}$ -or vice versa- and the frequency $\omega^{\text{sec}}_{+\frac{1}{2}, 1\leftrightarrow 0}$ or $\omega^{\text{sec}}_{-\frac{1}{2}, 1\leftrightarrow 0}$ with which the coherence evolves will change accordingly. At the time of the echo, the coherence will have gained a phase $\phi = \omega^{\text{sec}}_{\pm\frac{1}{2}\to\mp\frac{1}{2}, +1\leftrightarrow 0}t$, where $\omega^{\text{sec}}_{\pm\frac{1}{2}\to\mp\frac{1}{2}, +1\leftrightarrow 0}$ are the LiDEER frequencies of the two coherence transfer pathways:

$$\omega^{\text{sec}}_{+\frac{1}{2}\to-\frac{1}{2}, +1\leftrightarrow 0} = \left(E^{\text{sec}}_{\frac{1}{2},+1} - E^{\text{sec}}_{\frac{1}{2}, 0}\right) - \left(E^{\text{sec}}_{-\frac{1}{2}, +1} - E^{\text{sec}}_{-\frac{1}{2}, 0}\right) = \left(3\cos\left(\beta_{\text{dip}}\right)^2 - 1\right)\omega_{\text{dip}}, \tag{30}$$

$$\omega^{\text{sec}}_{-\frac{1}{2}\to+\frac{1}{2}, +1\leftrightarrow 0} = \left(E^{\text{sec}}_{-\frac{1}{2}, +1} - E^{\text{sec}}_{-\frac{1}{2}, 0}\right) - \left(E^{\text{sec}}_{\frac{1}{2}, 0} - E^{\text{sec}}_{\frac{1}{2}, -1}\right) = -\left(3\cos\left(\beta_{\text{dip}}\right)^2 - 1\right)\omega_{\text{dip}}. \tag{31}$$

When the other transition of the triplet spin from $m_T = 0$ and $m_T = -1$ is excited by the observer pulse, the frequencies are the same:

$$\omega^{\text{sec}}_{+\frac{1}{2}\to-\frac{1}{2}, 0\leftrightarrow -1} = \left(E^{\text{sec}}_{\frac{1}{2}, +1} - E^{\text{sec}}_{\frac{1}{2}, 0}\right) - \left(E^{\text{sec}}_{-\frac{1}{2}, +1} - E^{\text{sec}}_{-\frac{1}{2}, 0}\right) = \left(3\cos\left(\beta_{\text{dip}}\right)^2 - 1\right)\omega_{\text{dip}}, \tag{32}$$

$$\omega^{\text{sec}}_{-\frac{1}{2}\to+\frac{1}{2}, 0\leftrightarrow -1} = \left(E^{\text{sec}}_{-\frac{1}{2}, +1} - E^{\text{sec}}_{-\frac{1}{2}, 0}\right) - \left(E^{\text{sec}}_{\frac{1}{2}, 0} - E^{\text{sec}}_{\frac{1}{2}, -1}\right) = -\left(3\cos\left(\beta_{\text{dip}}\right)^2 - 1\right)\omega_{\text{dip}}. \tag{33}$$

As those are the same frequencies as the ones in DEER with two $S = 1/2$ spins, one eventually arrives at the same kernel $K_{\text{DEER}}(t, r)$. This means that like it was the case in LaserIMD the secular terms of the ZFS cancel each other out, and there is no effect of the ZFS on the LiDEER trace. In contrast to LaserIMD in secular approximation, there are also no coherence transfer pathways with $\Delta m_D = 0$, so that the maximum achievable modulation depth in LiDEER is 100 %.

It seems obvious that the same non-secular terms that lead to change in the LaserIMD frequencies are also relevant in LiDEER. Therefore, the LiDEER frequencies were also determined from the energy levels $E^{\text{non-sec}}_{m_D, m_T}$ that include the effects of the ZFS:

$$\begin{aligned}
\omega^{\text{non-sec}}_{+\frac{1}{2}\to-\frac{1}{2}, +1\leftrightarrow 0} &= \left(E^{\text{non-sec}}_{+\frac{1}{2}, +1} - E^{\text{non-sec}}_{+\frac{1}{2}, 0}\right) - \left(E^{\text{non-sec}}_{-\frac{1}{2}, +1} - E^{\text{non-sec}}_{-\frac{1}{2}, 0}\right) \\
&= \left(\left(3\cos\left(\beta_{\text{dip}}\right)^2 - 1\right) + 3\delta_{\text{ZFS}}\sin\left(2\beta_{\text{dip}}\right)\right)\omega_{\text{dip}},
\end{aligned} \tag{34}$$

$$\omega^{\text{non-sec}}_{-\frac{1}{2}\to+\frac{1}{2},\,+1\leftrightarrow 0} = \left(E^{\text{non-sec}}_{-\frac{1}{2},\,+1} - E^{\text{non-sec}}_{-\frac{1}{2},\,0}\right) - \left(E^{\text{non-sec}}_{+\frac{1}{2},\,0} - E^{\text{non-sec}}_{+\frac{1}{2},\,-1}\right)$$

$$= -\left(\left(3\cos(\beta_{\text{dip}})^2 - 1\right) + 3\delta_{\text{ZFS}}\sin(2\beta_{\text{dip}})\right)\omega_{\text{dip}}, \tag{35}$$

$$\omega^{\text{non-sec}}_{+\frac{1}{2}\to-\frac{1}{2},\,0\leftrightarrow -1} = \left(E^{\text{non-sec}}_{+\frac{1}{2},\,+1} - E^{\text{non-sec}}_{+\frac{1}{2},\,0}\right) - \left(E^{\text{non-sec}}_{-\frac{1}{2},\,+1} - E^{\text{non-sec}}_{-\frac{1}{2},\,0}\right)$$

$$= \left(\left(3\cos(\beta_{\text{dip}})^2 - 1\right) - 3\delta_{\text{ZFS}}\sin(2\beta_{\text{dip}})\right)\omega_{\text{dip}}, \tag{36}$$

$$\omega^{\text{non-sec}}_{-\frac{1}{2}\to+\frac{1}{2},\,0\leftrightarrow -1} = \left(E^{\text{non-sec}}_{-\frac{1}{2},\,+1} - E^{\text{non-sec}}_{-\frac{1}{2},\,0}\right) - \left(E^{\text{non-sec}}_{+\frac{1}{2},\,0} - E^{\text{non-sec}}_{+\frac{1}{2},\,-1}\right)$$

$$= -\left(\left(3\cos(\beta_{\text{dip}})^2 - 1\right) - 3\delta_{\text{ZFS}}\sin(2\beta_{\text{dip}})\right)\omega_{\text{dip}}. \tag{37}$$

It can be seen again that the ZFS leads to a shift in the dipolar frequencies. This shift is, besides the factor of 3, identical to the one that was obtained for the LaserIMD frequencies $\omega^{\text{non-sec}}_{+1}$ and $\omega^{\text{non-sec}}_{-1}$. From here, the next step is again the averaging over the orientations of the transient triplet label and the dipolar coupling vector that contribute to the LiDEER signal. However, this is even more complicated than it was in LaserIMD where all orientations are evenly excited by the laser-flash.

In LiDEER the triplet spins are also excited by microwave pulses which typically have a bandwidth that is much more narrow than the EPR spectrum of the transient triplet label. For example, the frequently used porphyrin labels have an EPR spectrum that is over 2 GHz broad (Di Valentin et al., 2014) of which a typical rectangular microwave pulse with a length of 10 ns can only excite roughly 120 MHz (Schweiger and Jeschke, 2001). Therefore, not all orientations of the transient triplet labels contribute to the LiDEER signal and it is rather tedious to even derive an expression for the integrals that describe the

orientation averaging. To circumvent this problem, the LiDEER traces will be calculated by time-domain simulations with weak microwave pulses in the next sections.

## 3 Materials and methods

### 3.1 Simulations

The powder averages for LaserIMD were performed by a numerical integration of Eq. (25) -(27) with home-written MATLAB

(version 2020b) scripts. For the angle $\beta_{\text{dip}}$ a linear, equidistant grid from 0 to $\frac{\pi}{2}$ was used. Each value was weighted proportional to $\sin(\beta_{\text{dip}})$. For the orientation of the transient triplet label, a grid with all three Euler angles $\alpha_{\text{T}}$, $\beta_{\text{T}}$ and $\gamma_{\text{T}}$, including the corresponding weights, was calculated according to the REPULSION approach (Bak and Nielsen, 1997; Hogben et al., 2011) with the software package *Spinach* version 2.6.5625 (Hogben et al., 2011). To check for a sufficient convergence, a test run with an increasing numbers of points for the two grids was simulated. The test run was stopped when the relative

change $\Delta\epsilon$ in the simulated signal, when the number of grids points was increased, was below 1 %. For $\beta_{\text{dip}}$ a grid size of 200

points was sufficient, whereas for $\alpha_\mathrm{T}$, $\beta_\mathrm{T}$ and $\gamma_\mathrm{T}$ 12800 points were necessary. For details of the convergence behavior, see S3.

The time-domain simulations for LiDEER were performed with *Spinach* version 2.6.5625 (Hogben et al., 2011). The powder averaging was done with the same grids that were used for LaserIMD. For details see S8. The source code for the LiDEER

simulations can be downloaded at https://github.com/andreas-scherer/LiDEER_simulations.git

### 3.2 Experiments and data analysis

LaserIMD and LiDEER measurements were performed on the two peptides TPP-pAA$_5$-NO• and TPP-pAA$_{10}$-NO• shown in Figure 3. They were purchased from Biosynthan (Berlin) as powder samples and used without further purification. They were dissolved in MeOD/D$_2$O (98/2 vol.%) and prior to freezing in liquid nitrogen, they were degassed with three freeze-pump-

thaw cycles. Light excitation was performed at a wavelength of 510 nm by an Nd:YAG laser system from Ekspla (Vilnius) that was coupled into the resonator via a laser fiber. EPR measurements were performed on a commercial Bruker E580 spectrometer, X-band measurements in an ER4118X-MS3 resonator and Q-band measurements in an ER5106QT-2 resonator. In X-band the resonator was critically coupled to a Q-value of ≈ 900-2000and in Q-band it was overcoupled to a Q-value of ≈ 200. LaserIMD was recorded with the pulse sequence $\pi/2 - \tau - \pi - $ t - laser pulse - (τ-t) - echo (Hintze et al., 2016). A 2-step

phase cycle was implemented for baseline correction. Signal averaging was done by recording 10 shots per point. The zero-time correction was performed by recording a short reLaserIMD (Dal Farra et al., 2019a) trace as reported in (Scherer et al., 2022). LiDEER measurements were performed with the pulse sequence: laser pulse $- $ DAF $- \pi/2 - \tau_1 - \pi - $ t $- \pi_\mathrm{pump} - (\tau_1 + \tau_2 - $ t$) - \pi - \tau_2 - $ echo (Di Valentin et al., 2014). The delay-after-flash (DAF) was set to 500 ns and $\tau_1$ to 400 ns. Nuclear modulation averaging was performed by varying the $\tau_1$ time in 8 steps with $\Delta\tau_1 = 16$ ns. Phase cycling was performed with an

8-step scheme ((x) [x] xp x) as proposed by (Tait and Stoll, 2016). The LiDEER data were analysed with the python software package *DeerLab* (version 0.13.2) (Fábregas Ibáñez et al., 2020) and *Python* 3.9 with the DEER kernel $K_\mathrm{DEER}(t,r)$ and Tikhonov regularization. A 3D homogenous background function was used and the regularization parameter was chosen according to the Akaiki information criterion (Edwards and Stoll, 2018). The validation was performed with bootstrapping by analyzing 1000 samples that are generated with artificial noise. The error was then calculated as the 95% confidence interval.

Further details can be found in S7 and S10.

**Figure 3:** Chemical structures of the peptides TPP-pAA$_5$-NO• and TPP-pAA$_{10}$-NO• with the letter code Ala: L-alanine and Aib: α-isobutyric acid.

## 4 Results and Discussion

### 4.1 LaserIMD simulations

An initial simulation to study the effect of the ZFS in LaserIMD was performed for X-band ($\nu_\mathrm{T} = 9.3\,\mathrm{GHz}$) with a dipolar coupling that corresponds to a distance of $r = 2.2\,\mathrm{nm}$, a ZFS of $D = 1159\,\mathrm{MHz}$ and $E = -238\,\mathrm{MHz}$ and zero-field

populations $P_\mathrm{x} = 0.33$, $P_\mathrm{Y} = 0.41$ and $P_\mathrm{Z} = 0.26$. The ZFS and zero-field populations correspond to TPP that is often used to perform LaserIMD and LiDEER measurements (Di Valentin et al., 2014; Hintze et al., 2016; Di Valentin et al., 2016; Bieber et al., 2018; Bertran et al., 2020). For simplicity, a complete excitation of the transient triplet label ($\gamma = 1$) was assumed and no background was added ($B(t) = 1$). For a more detailed analysis, the contributions from the three coherence transfer pathways with $\Delta m_\mathrm{T} = 1, 0, -1$, termed $V_{+1}^{\mathrm{non-sec}}(t)$, $V_0^{\mathrm{non-sec}}(t)$ and $V_{-1}^{\mathrm{non-sec}}(t)$, are simulated separately and presented in

Figure 4 together with their resulting sum $V_{\mathrm{LaserIMD}}^{\mathrm{non-sec}}(t)$. They are also compared with the corresponding traces from the secular approximation $V_{\mathrm{LaserIMD}}^{\mathrm{sec}}(t)$, $V_{+1}^{\mathrm{sec}}(t)$, $V_0^{\mathrm{sec}}(t)$ and $V_{-1}^{\mathrm{sec}}(t)$, where the ZFS is ignored. The comparison of the traces including and excluding the ZFS ($V_{+1}^{\mathrm{non-sec}}(t)$ and $V_{-1}^{\mathrm{non-sec}}(t)$ with $V_{+1}^{\mathrm{sec}}(t)$ and $V_{-1}^{\mathrm{sec}}(t)$) in Figure 4a and c shows that there is no visible effect of the ZFS in the traces $V_{+1}^{\mathrm{non-sec}}(t)$ and $V_{-1}^{\mathrm{non-sec}}(t)$ and they look virtually identical to $V_{+1}^{\mathrm{sec}}(t)$ and $V_{-1}^{\mathrm{sec}}(t)$. The frequency shift $\delta_{\mathrm{ZFS}}\sin(2\beta_{\mathrm{dip}})\omega_{\mathrm{dip}}$ seems to be averaged out after integration for these terms. The situation is different

in the case of $V_0^{\mathrm{non-sec}}(t)$ and $V_0^{\mathrm{sec}}(t)$ in Figure 4b. Whereas $V_0^{\mathrm{sec}}(t)$ is a constant function of time and does not contribute to the echo modulation, $V_0^{\mathrm{non-sec}}(t)$ shows a continuous decay of the echo intensity with increasing time. This decay does not contain any additional dipolar oscillations and its shape does not seem to follow any obvious simple mathematical law. For the full LaserIMD traces in Figure 4d, this means that, whereas without taking ZFS into account the trace $V_{\mathrm{LaserIMD}}^{\mathrm{sec}}(t)$ looks like a $S = 1/2$ DEER trace with and a modulation depth of $\lambda = 66.\bar{6}\,\%$. , the trace $V_{\mathrm{LaserIMD}}^{\mathrm{non-sec}}(t)$ with the ZFS shows the same

dipolar oscillations but on top of a decay. This also means that, due to the coherence transfer pathway with $\Delta m_\mathrm{T} = 0$ also resulting in a variation of the echo intensity, the modulation depth of LaserIMD is increased by the ZFS and values higher than $66.\bar{6}\,\%$. can be reached.

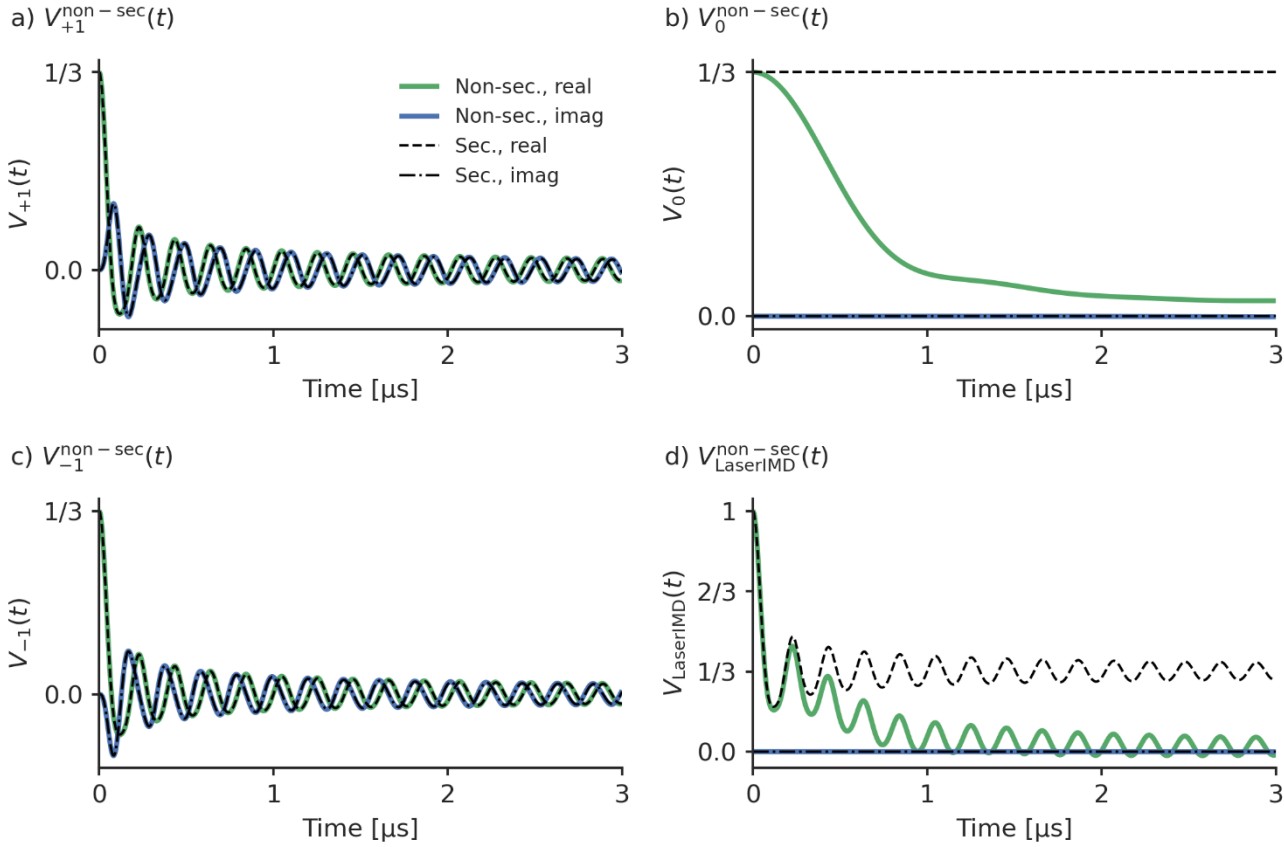

**Figure 4:** Comparison of simulated LaserIMD traces with and without non-secular interactions with the values $D = 1159$ MHz, E= $-238$ MHz and $P_x = 0.33$, $P_y = 0.41$ and $P_z = 0.26$, $\nu_T = 9.3$ GHz (X-band) and $r = 2.2$ nm. a) $V_{+1}^{non-sec}(t)$, b) $V_0^{non-sec}(t)$, c) $V_{-1}^{non-sec}(t)$, d) $V_{LaserIMD}^{non-sec}(t) = V_{+1}^{non-sec}(t) + V_0^{non-sec}(t) + V_{-1}^{non-sec}(t)$.

5    The frequency shift caused by the non-secular terms of the ZFS in LaserIMD depends not only on $D$ and $E$, but also on the zero-field populations $P_x$, $P_y$ and $P_z$, the Zeeman frequency $\nu_T$ and the distance $r$ (see Eq. (21) -(24)). The influence of these parameters was studied by simulating additional LaserIMD traces with different magnetic field strengths, ZFS values, zero-field populations, and distance distributions (see Figure 5 and Figure 6). In Figure 5a, two LaserIMD traces in X- and Q-band ($\nu_T = 9.3$ GHz and $\nu_T = 34.0$ GHz) with TPP as a transient triplet label and a distance of $r = 2.2$ nm are compared. Figure 10  5b shows the comparison between the ZFS of TPP ($D = 1159$ MHz and E= $-238$ MHz) and a stronger ZFS of $D = 3500$ MHz and E= $-800$ MHz, as such high values are possible for some labels like Rose Bengal and Erythrosin B (Williams et al., 2020; Bertran et al., 2022b). Both simulations were performed in Q-band with $r = 2.2$ nm. Figure 5c shows three simulations with the population of the zero-field triplet states being completely assigned to either $P_x$, $P_y$ or $P_z$. In Figure 5d, the effect of different distances of $r = 2.2$ nm and $r = 5$ nm on $V_0^{non-sec}(t)$ is shown for TPP in Q-band. The simulations in 15  Figure 5 were all done with a single distance. To study the influence of the width of the distance distribution on $V_0^{non-sec}(t)$,

additional simulations were performed with a Gaussian distance distribution with a mean of 3 nm and different standard deviations $\sigma$ ranging from 0.05 nm to 3 nm. The results of these simulations are shown in Figure 6a (X-band) and Figure 6b (Q-band).

Figure 5a, Figure 5b and Figure 5c show that there are no visible differences in the dipolar oscillations in $V_{+1}^{\text{non-sec}}(t)$ and $V_{-1}^{\text{non-sec}}(t)$, when the Zeeman frequency, ZFS or zero-field populations are changed. This can also be seen in the SI in S4, S5 and S6 where the traces for different Zeeman frequencies, ZFSs and distances are compared in more detail. This agrees with the former results in Figure 4 that the frequency shift due to the ZFS is virtually averaged out in a powder sample for $V_{+1}^{\text{non-sec}}(t)$ and $V_{-1}^{\text{non-sec}}(t)$, so changing the involved parameters should also have little effect. The situation is different for $V_{0}^{\text{non-sec}}(t)$, which, as it is shown in Figure 4c, is more strongly affected by of the ZFS. The previously mentioned decay is faster for lower Zeeman frequencies (see Figure 5a) and a stronger ZFS (see Figure 5b). Because $\delta_{\text{ZFS}}$ ultimately depends on the ratio of the ZFS to the Zeeman frequency, a higher ZFS and a lower Zeeman frequency both increase the magnitude of the frequency shift of $\omega_{0}^{\text{non-sec}}$ in the same way which leads to the same effect on the LaserIMD trace. The parameters that have the least influence on the LaserIMD trace are the zero-field populations (see Figure 5c). Changing the populations of the zero-field states does not seem to affect the dipolar oscillations, as it was the case for different ZFSs and magnetic field strengths. This time, also the decay of $V_{0}^{\text{non-sec}}(t)$ is barely affected by different zero-field populations. Figure 5d shows that shorter distances lead to a faster decay of $V_{0}^{\text{non-sec}}(t)$. As can be seen in Eq. (21) -(23), changing the distance $r$ from 2.2 to 5 nm leads to an increase of the LaserIMD frequencies $\omega_{+1}^{\text{non-sec}}$, $\omega_{0}^{\text{non-sec}}$ and $\omega_{-1}^{\text{non-sec}}$ that scales with $r^{-3}$. This distance dependence of the dipolar oscillations (not shown in Figure 5c) is used in PDS for the calculation of the distance distributions. In the case of LaserIMD, the steepness of the decay of $V_{0}^{\text{non-sec}}(t)$ is an additional feature that depends on the distance between the spin labels. As can be seen in Figure 6 the width of the distance distribution also has an influence on the decay of $V_{0}^{\text{non-sec}}(t)$. In X-band (see Figure 6a) and for small standard deviations of $\sigma = 0.05$ nm, $V_{0}^{\text{non-sec}}(t)$ has a sigmoid like shape. Increasing the width has a twofold effect on the decay of $V_{0}^{\text{non-sec}}(t)$. Whereas the initial decay is steeper, on a long scale, the decay of $V_{0}^{\text{non-sec}}(t)$ is decreased for broader distance distributions. This can clearly be seen in the case of $\sigma = 3$ nm where for $t < 1$ μs $V_{0}^{\text{non-sec}}(t)$ decays faster for the simulation with $\sigma = 3$ nm than with $\sigma = 0.05$ nm, but for $t > 1$ μs $V_{0}^{\text{non-sec}}(t)$ decays slower for $\sigma = 3$ nm than for $\sigma = 0.05$ nm. In Q-band where the decay of $V_{0}^{\text{non-sec}}(t)$ is generally slower, the simulations in Figure 6b show that here only the first effect is of relevance. It can be seen that the first part of the decay of $V_{0}^{\text{non-sec}}(t)$ is again steeper for broader distance distributions, but the second part where this behavior is inverted lies outside the time window. This means that in Q-band the width of the distance distribution has a smaller influence on the decay of $V_{0}^{\text{non-sec}}(t)$ than in Q-band.

Taken together, variation in the ZFS parameter, the population of the ZFS states and the employed magnetic field (X- or Q-band) do not affect the dipolar oscillations in $V_{+1}^{\text{non-sec}}(t)$ and $V_{-1}^{\text{non-sec}}(t)$. They mostly have an effect on the decay of $V_{0}^{\text{non-sec}}(t)$, such that larger ZFS parameters and lower magnetic fields will lead to a stronger additional decay in the LaserIMD trace. The additional decay also depends on the distance distribution between the spin labels, it is faster for shorter

distances and the shape of the decay also depends on the width of the distance distribution (in X-band more than in Q-band). The decay of $V_0^{non-sec}(t)$ can therefore be used as an additional source of information for the calculation of the distance distribution.

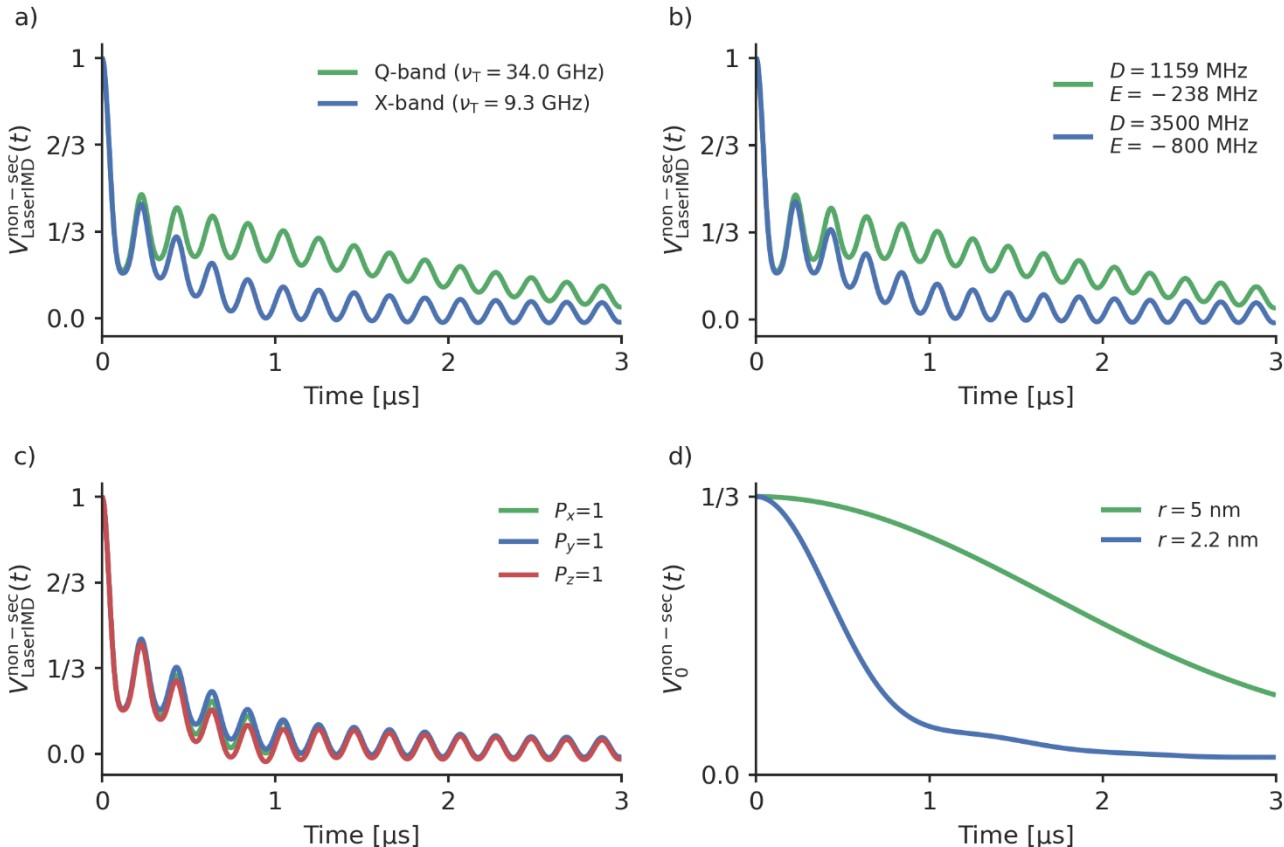

**Figure 5:** A comparison of different LaserIMD traces $V_{LaserIMD}^{non-sec}(t)$ with different parameters. The following values were used for the simulations. a) TPP, $r = 2.2$ nm and $\nu_T = 34$ GHz (green) and $\nu_T = 9.3$ GHz (blue) b) $P_x = 0.33$, $P_y = 0.41$, $P_z = 0.26$, $r = 2.2$ nm, $\omega_T = 9.3$ GHz and $D = 1159$ MHz, E= −238 MHz (green) and $D = 3500$ MHz, E= −800 MHz (blue)   c) TPP, $\nu_T = 34$ GHz  and $r = 2.2$ nm green) and $r = 5$ nm (blue) d) $D = 1159$ MHz, E= −238 MHz $r = 2.2$ nm, $\nu_T = 9.3$ GHz and $P_x = 1$, $P_y = 0$, $P_z = 0$ (green), $P_x = 0$, $P_y = 1$, $P_z = 0$ (blue) and $P_x = 0$, $P_y = 0$, $P_z = 1$ (red).

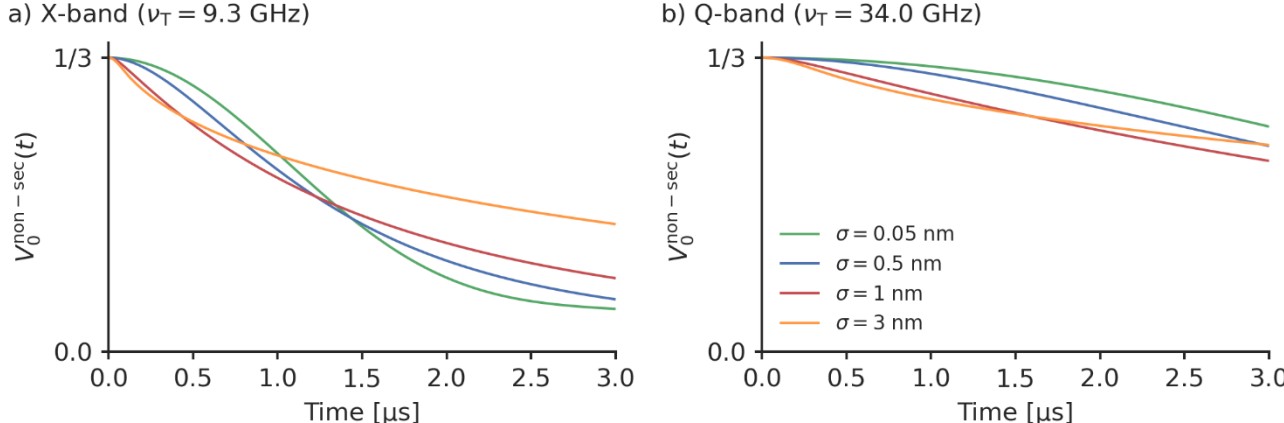

**Figure 6:** The influence of the width of the distance distribution on the decay of $V_0^{non-sec}(t)$ for TPP in a) X-band and b) Q-band. The simulations were performed for a Gaussian distance distribution width a mean of 3 nm and different standard deviations $\sigma$.

So far, all simulations only showed a visible effect of the ZFS on $V_0^{non-sec}(t)$, no significant influence on $V_{+1}^{non-sec}(t)$ and $V_{-1}^{non-sec}(t)$ was observed. To check if and when the ZFS has also an influence on $V_{+1}^{non-sec}(t)$ and $V_{-1}^{non-sec}(t)$, we performed additional simulations where the effect of the ZFS is expected to be stronger. This can be obtained by either lower Zeeman frequencies or higher ZFS values. As the effect on $\delta_{ZFS}$ is the same in both cases, the ratio of $D$ and the Zeeman frequency of the triplet $\nu_T$ can be defined as:

$$q = \frac{D}{2\pi\nu_T} \tag{38}$$

For simplification, the ZFS was assumed to be axial with $E = 0$. This simplifies the expression of $\delta_{ZFS}$ to:

$$\delta_{ZFS} = \frac{3}{4} q \sin(2\beta_T) \cos(\alpha_T) \tag{39}$$

The simulation in X-band with TPP from Figure 4 corresponds to a ratio where $q$ is approximately 0.13. Here, we tried values for $q$ of up to 1. Figure 7 shows the sum of $V_{+1}^{non-sec}(t)$ and $V_{-1}^{non-sec}(t)$ of these simulations and compares it to a trace where the effect of the ZFS has been ignored. It can be seen that up to $q = 0.5$, the traces are negligibly affected by the ZFS. For higher values, the dipolar oscillations start to get shifted to slightly higher frequencies and are also smoothed out more quickly. Analyzed with the over-simplified kernel $K_{DEER}(t,r)$ of the $S = 1/2$ model, this would result in a shift to smaller distances and an artificial broadening of the distance distribution. However, for experimentally relevant distance distributions with a finite width, the oscillations typically fade out much quicker and cases where four oscillations can be resolved are scarce. In such a case, the observed influence of the ZFS for high values of $q$ can be expected to almost negligible. Furthermore, as $q = 1$ is equivalent to a ZFS that is in the same order of magnitude as the Zeeman frequency, this is not relevant for most practical applications, as LaserIMD is typically performed at X- or Q-band ($\nu_T = 9.3$ GHz or $\nu_T = 34.0$ GHz) and all transient triplet labels used so far have a ZFS value $D$ below 4 GHz (Dal Farra et al., 2019b; Williams et al., 2020). Even in the most extreme

case, this would result in values for $q$ smaller than 0.5. Consequently, the effect of the ZFS on $V_{+1}^{\text{non-sec}}(t)$ and $V_{-1}^{\text{non-sec}}(t)$ is not relevant for most experiments and even though the $V_{+1}^{\text{non-sec}}(t)$ and $V_{-1}^{\text{non-sec}}(t)$ can in principle be influenced by the ZFS, it seems to be a safe assumption that the ZFS in LaserIMD affects only the decay in $V_0^{\text{non-sec}}(t)$ and not the dipolar oscillations in $V_{+1}^{\text{non-sec}}(t)$ and $V_{-1}^{\text{non-sec}}(t)$.

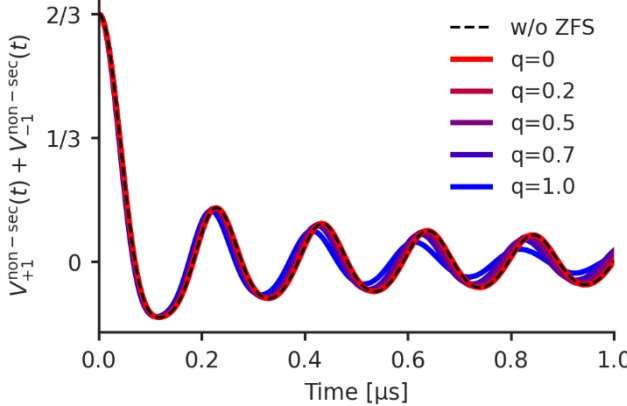

**Figure 7:** a) The sum of $V_{+1}^{\text{non-sec}}(t)$ and $V_{-1}^{\text{non-sec}}(t)$ for different values of $q$ and $P_x = 0.33$, $P_y = 0.41$, $P_z = 0.26$ and $r = 2.2$ nm. Only the real part is shown.

As was stated before, in the secular approximation, LaserIMD traces can be analysed with the kernel $K_{\text{DEER}}(t, r)$ of the $S = 1/2$ model. To check to what extent this is true when the ZFS is not negligible, we simulated LaserIMD traces that were subsequently analyzed with $K_{\text{DEER}}(t, r)$. To mimic experimental conditions more closely, we assumed an incomplete excitation of the transient triplet label and the intermolecular dipolar background was also considered. TPP was used as transient triplet label with a distance to the permanent spin label of $r = 2.2$ nm and a modulation depth of $\lambda = 50$ %, which roughly correspond to the values that can be typically achieved in experiments. Simulations were performed in X- and Q-band with different background decay rates varying between $k = 0.0$ µs$^{-1}$ (no background) to $k = 0.4$ µs$^{-1}$. The resulting traces were then analyzed with $K_{\text{DEER}}(t, r)$ and Tikhonov regularization (see details in S7).

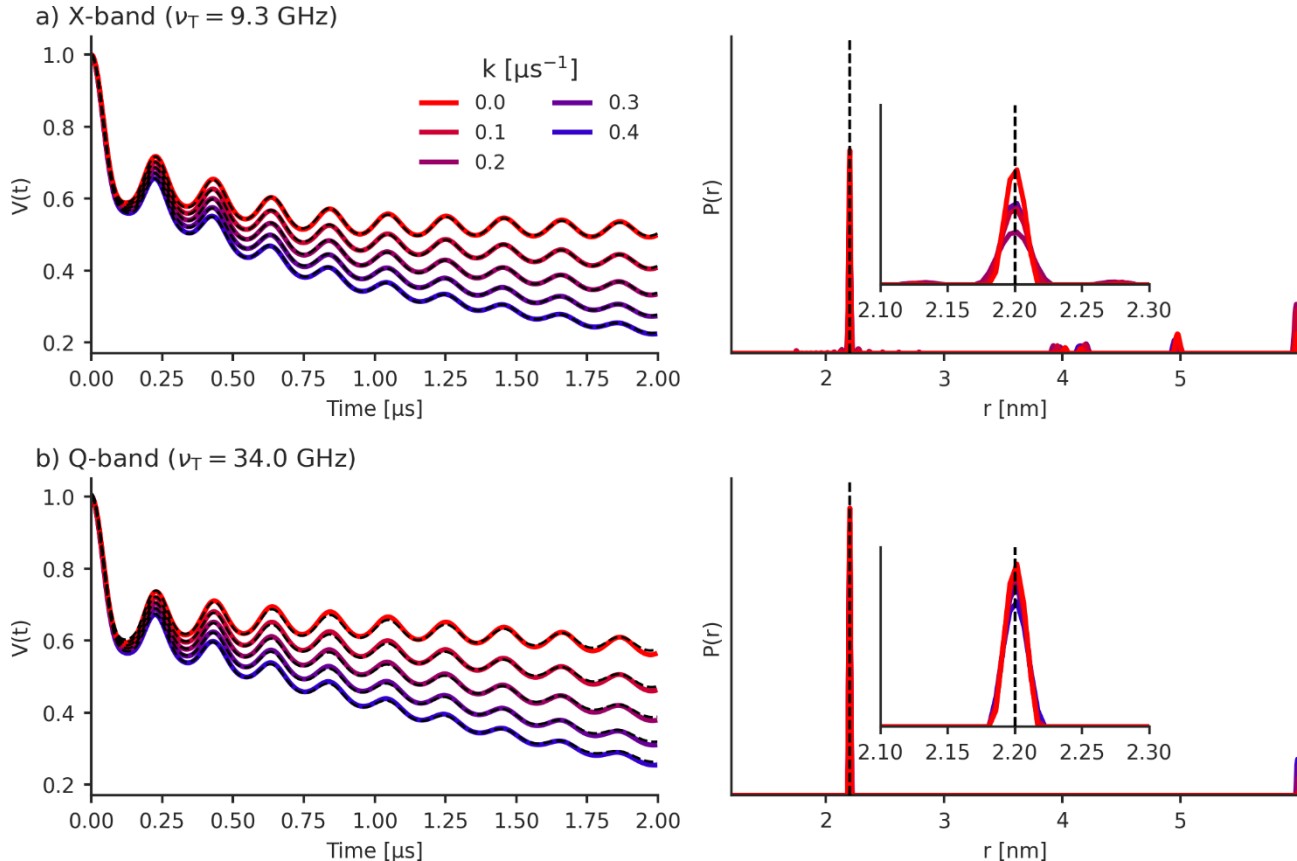

**Figure 8:** Simulationed LaserIMD traces $V_{\text{LaserIMD}}^{\text{non-sec}}(t)$ including the ZFS for TPP as transient triplet label and $r = 2.2$ nm in a) X-band ($\nu_T = 9.3$ GHz) and b) Q-band ($\nu_T = 34.0$ GHz). The background decay that was used for the simulation was varied between $k = 0.0$ μs$^{-1}$ and $k = 0.4$ μs$^{-1}$. The left side shows the simulated traces (with the fits as dashed black line) and the right side shows the distance distributions that were obtained with Tikhonov regularization with $K_{\text{DEER}}(t, r)$. The true distance of $r = 2.2$ nm is plotted as dashed black line.

**Table 1:** The background decay values and modulation depths that were determined for the simulations from Figure 8. The modulation depth for the simulations was always set to $\lambda = 50\ \%$.

| $k\ [\mu s^{-1}]$ | X-band ($\nu_T = 9.3$ GHz) | | Q-band ($\nu_T = 34.0$ GHz) | |
|---|---|---|---|---|
| | $k_{\text{fit}}\ [\mu s^{-1}]$ | $\lambda_{\text{fit}}\ [\%]$ | $k_{\text{fit}}\ [\mu s^{-1}]$ | $\lambda_{\text{fit}}\ [\%]$ |
| **0.0** | 0.00 | 47 | 0.07 | 32 |
| **0.1** | 0.00 | 54 | 0.17 | 32 |
| **0.2** | 0.00 | 61 | 0.26 | 33 |
| **0.3** | 0.00 | 66 | 0.35 | 34 |
| **0.4** | 0.01 | 70 | 0.44 | 36 |

The simulations and fitted distance distributions can be seen in Figure 8 and the background decay rates and modulations depths that were obtained by the fits in Table 1. Figure 8 shows that the fits agree well with the simulated data and the main peak of the distance distribution at $r = 2.2$ nm is fitted appropriately in X- as well as in Q-band. However, there can be additional artifact peaks in the distance distributions, and the fitted modulation depths and background decay rates can be erroneous (see Table 1). This is particularly pronounced in X-band, which shows artifacts in the distance distribution between 3.9 nm and 5 nm and at the higher distance end. Moreover, the background decay rates and modulation depths deviate significantly from the values that were originally used for the simulations. The simulations in X-band are always fitted with a background decay rate close to zero ($k_{\text{fit}} \approx 0.0\ \mu s^{-1}$), even in the cases where the strongest background was included ($k = 0.4\ \mu s^{-1}$) in the simulation. The modulation depth was fitted with values from 47 % to 70 % and varies significantly for different background decays. In Q-band, the fitted parameters are closer to the input values of the simulations. The distance artifacts that appeared in X-band between 3.9 nm and 5 nm have disappeared, and only those at the long distance limit remain. In Q-band the fitted background decay is always a bit larger than the true value. Except for the case were the true background decay is set to $k = 0\ \mu s^{-1}$, the deviation of the fitted and the true background decay is smaller in Q-band than in X-band.. Only the obtained modulation depths are less accurate than in X-band and fitted to values between 32 % and 36 %. Even though, these simulations are only anecdotal evidence and generalizations from these data must be taken with caution, they show that when LaserIMD data are analyzed with $K_{\text{DEER}}(t,r)$ it is possible to extract the main distance peak correctly. Analyzing LaserIMD traces with $K_{\text{DEER}}(t,r)$ can thus be an option in situations where the ZFS values and zero-field populations of the transient triplet label are unknown and their effect cannot be included in the analysis. However, this way of analyzing LaserIMD data can give artifacts at higher distances and also errors in the obtained modulation depth and background decay rate. This is particularly pronounced for low magnetic fields (e.g. X-band) and similar results can be expected for transient triplet labels with higher ZFS values.

## 4.2 LiDEER simulations

In LaserIMD, transient triplet labels of all orientations are excited by the laser flash and contribute to the signal, thus an integration over all orientations was performed (Eq. (25) -(27)) to calculate the LaserIMD signal. Contrary to that, in LiDEER the transient triplet labels are additionally excited by microwave observer pulses. As the spectrum of many transient triplet labels exceeds the excitation bandwidth of these pulses (Di Valentin et al., 2014; Williams et al., 2020; Krumkacheva et al., 2019), only a small number of orientations within the excitation bandwidth contribute to the signal. Because the frequency shift $\delta_{ZFS}$ of the LiDEER frequencies (Eq. (34) -(37)) depends on the orientation of the transient triplet labels, the choice of the observer frequency influences the shape of the LiDEER trace.

In experiments, when the commonly used nitroxides or other spin labels with $g_D \approx 2$ are used as pump spin, the resonator bandwidth allows to use only the $Y^{\pm}$ peaks as observer position, because the other parts of the EPR spectrum of the transient triplet label lie outside the resonator bandwidth (Bieber et al., 2018; Bowen et al., 2021). Figure 9 shows the orientations of the triplet label TPP that in this case contribute to the LiDEER signal. The contribution of the orientations where the Y axis of eigenframe of the ZFS is parallel to the external magnetic field ($\beta_T = \pi/2$ and $\gamma_T = \pi/2$) is eponymous for the $Y^{\pm}$ peaks. For this orientation, the frequency shift $\delta_{ZFS} = 0$ and the ZFS has no effect on the LiDEER trace. However, it can be seen that other orientations are also excited if the observer pulses are placed on either of the $Y^{\pm}$ peaks. For these contributions it cannot guaranteed that $\delta_{ZFS}$ is always zero, so that there might still be an effect of the ZFS.

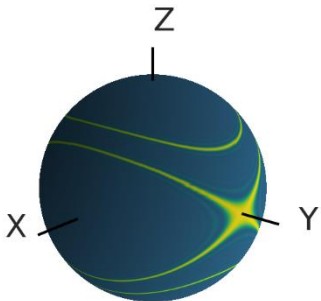

**Figure 9:** The orientations (shown in yellow) of the transient triplet label that are excited by a rectangular $\pi$ pulse with a pulse length of 20 ns that is placed on the $Y^+$ peak of EPR spectrum of TPP in Q-band. For the calculation, the magnetic field was set to $B = 1.2097$ T and the pulse frequency was set to 33.646 GHz. The position of the pulse relative to the EPR spectrum is shown in Fig. S7. The angle $\beta_T$ is the polar angle of the depicted sphere and the angle $\gamma_T$ is the azimuthal angle.

To study the effect of the ZFS in LiDEER, numerical time-domain simulations for different ZFS values in X- and Q-band were performed. The microwave pulses were placed on the $Y^+$ peak of the EPR spectrum and had a finite length, power and bandwidth so that only the orientations that are shown in Figure 9 contribute to the LiDEER signal, as it is the case in the experimental setup. A simulation for TPP as transient triplet label was performed in X- and Q-band and an additional

simulation with a larger ZFS of $D = 3500$ MHz and $E = -800$ MHz was performed in X-band. The permanent spin label was included as a doublet spin with an isotropic $g$-value ($g_D = 2$) and without any additional hyperfine interactions. The distance was set to $r = 2.2$ nm and no background from intermolecular spins was included. To check for artifacts that occur in distance distributions if the ZFS is ignored in data analysis, the simulated LiDEER traces were analyzed with $K_{DEER}(t, r)$ and Tikhonov regularization. The details of the calculation of the distance distribution in S7 and the details of the simulations can be found in S8.

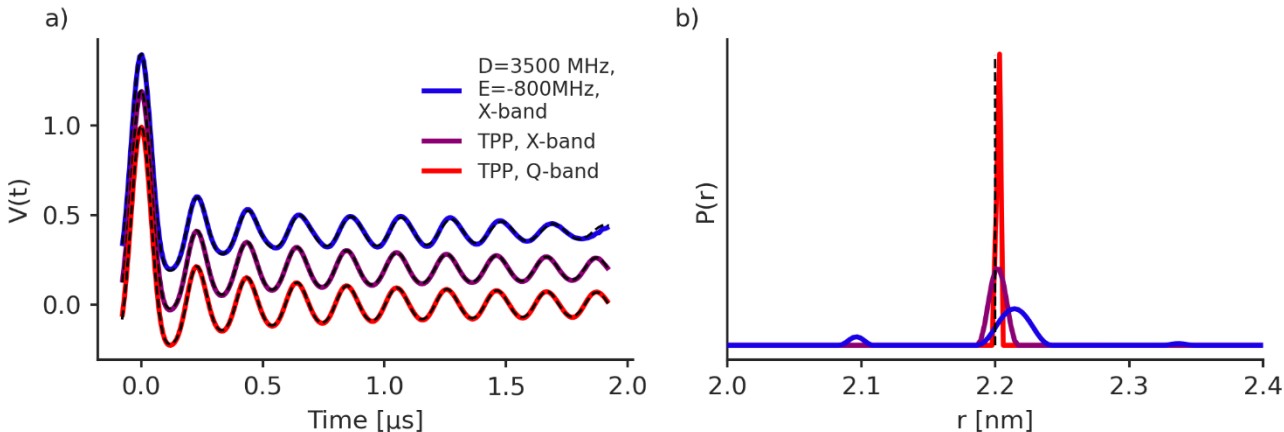

**Figure 10:** a) LiDEER simulations with the observer pulse placed on the Y⁺-peak of the EPR spectrum of the transient triplet label in different frequency bands and with different ZFS. The traces are shifted by 0.2 for better visibility. For Q-band and TPP, the magnetic field was set to 1.2097 T and the observer frequency to 33.646 GHz; for X-band, the magnetic field was set to 0.33 T, which for TPP corresponds to an observer frequency of 9.042 GHz and for a ZFS with $D = 3500$ MHz and $E = -800$ MHz to an observer frequency of 9.042 GHz. The position of the observer and pump pulse with respect to the EPR spectrum is shown in Fig. S7a, c and e. The further parameters were $P_x = 0.33$, $P_y = 0.41$, $P_z = 0.26$ and $r = 2.2$ nm. The numerical simulations were fitted with Tikhonov regularization. The fits are shown as dashed black lines. b) The corresponding distance distribution. The true distance of $r = 2.2$ nm is plotted as dashed black line.

Figure 10a shows the simulated LiDEER traces and Figure 10b the obtained distance distributions. The differences in the LiDEER traces for different ZFS and Zeeman frequencies are smaller than they are in LaserIMD (see Figure 4). This is because in LiDEER, there is no equivalence for the coherence transfer pathway with $\Delta m_T = 0$ that showed the strongest dependency on the ZFS and magnetic fields in LaserIMD (see Figure 5). The distance distribution for TPP in Q-band shows a narrow peak at 2.20 nm with a full width at half maximum (FWHM) of 0.004 nm. This fits to the 2.20 nm (FWHM $= 0$ nm) that were used for the simulation. In X-band, the distance distribution with TPP is also centered at 2.20 nm, but gets broadened to a FWHM of 0.014 nm. This trend increases for the large ZFS with $D = 3500$ MHz and $E = -800$ MHz in X-band. Here, the distance distribution gets even broader with an FWHM of 0.028 nm and is now also shifted to a center of $\approx 2.22$ nm. This behavior fits to the results of LaserIMD in Figure 7, where the shifts of the dipolar oscillation get also larger when the ZFS is large compared to the Zeeman frequency. However, it must also be stated that the observed shifts of the distance distribution are still rather small here and should be below the resolution limit that is relevant in most experiments. Additional traces, where the observer pulse is set off-resonance to the canonical peaks were also performed and are presented in S9. Here, the effect of

the ZFS can clearly be seen and the LiDEER trace of the simulation with $D = 3500$ MHz and $E = -800$ MHz in X-band shows strong deviations from the other traces that were simulated with a smaller ZFS. The dipolar oscillations fade out much faster, which also leads to a stronger broadening of the distance distributions. However, for experimentally relevant cases with distance distributions of a finite width, the oscillations in the dipolar trace fade out much faster anyway. It is to be expected

that in these cases, the effect of the ZFS on the LiDEER trace are rather small and that therefore artifacts in the distance distribution are not so pronounced, even in the case when the observer pulses are set to a non-canonical orientation.

This means that in general the ZFS has an effect on LiDEER and the LiDEER trace changes, when different parts of the EPR-spectrum of the transient triplet label are used for excitation by the observer pulses. However, in the special case when either of the $Y^{\pm}$ peaks is used as position for the observer pulse, the effect of the ZFS can be suppressed and LiDEER traces can be

analyzed with the $K_{\mathrm{DEER}}(t, r)$ kernel without introducing significant artfacts in the distance distribution. This is particularly valid for TPP -and other transient triplet labels with a similar ZFS- in Q-band.

## 4.3 Experiments

To experimentally confirm the theoretical finding that the ZFS has an influence on the shape of the LaserIMD trace, LaserIMD measurements were performed at different magnetic field strengths at X- and Q-band and with two model systems with a

shorter and longer distances between the labels. This should result in scenarios were the ZFS has either a weak effect on the trace (high magnetic field strength and long distance) or strong effect (low magnetic field strength and short distance). The LaserIMD experiments were simulated with the newly derived model that includes the ZFS. The distance distributions and background decay rates that were used for these simulations of the LaserIMD traces were determined with LiDEER. The measurements were performed with the peptides TPP-pAA$_5$-NO• and TPP-pAA$_{10}$-NO•. They contain TPP as transient triplet

label and the nitroxide TOAC as permanent spin label. Both labels are separated by a rather rigid helix consisting of L-alanine and α-isobutyric acid (Di Valentin et al., 2016).

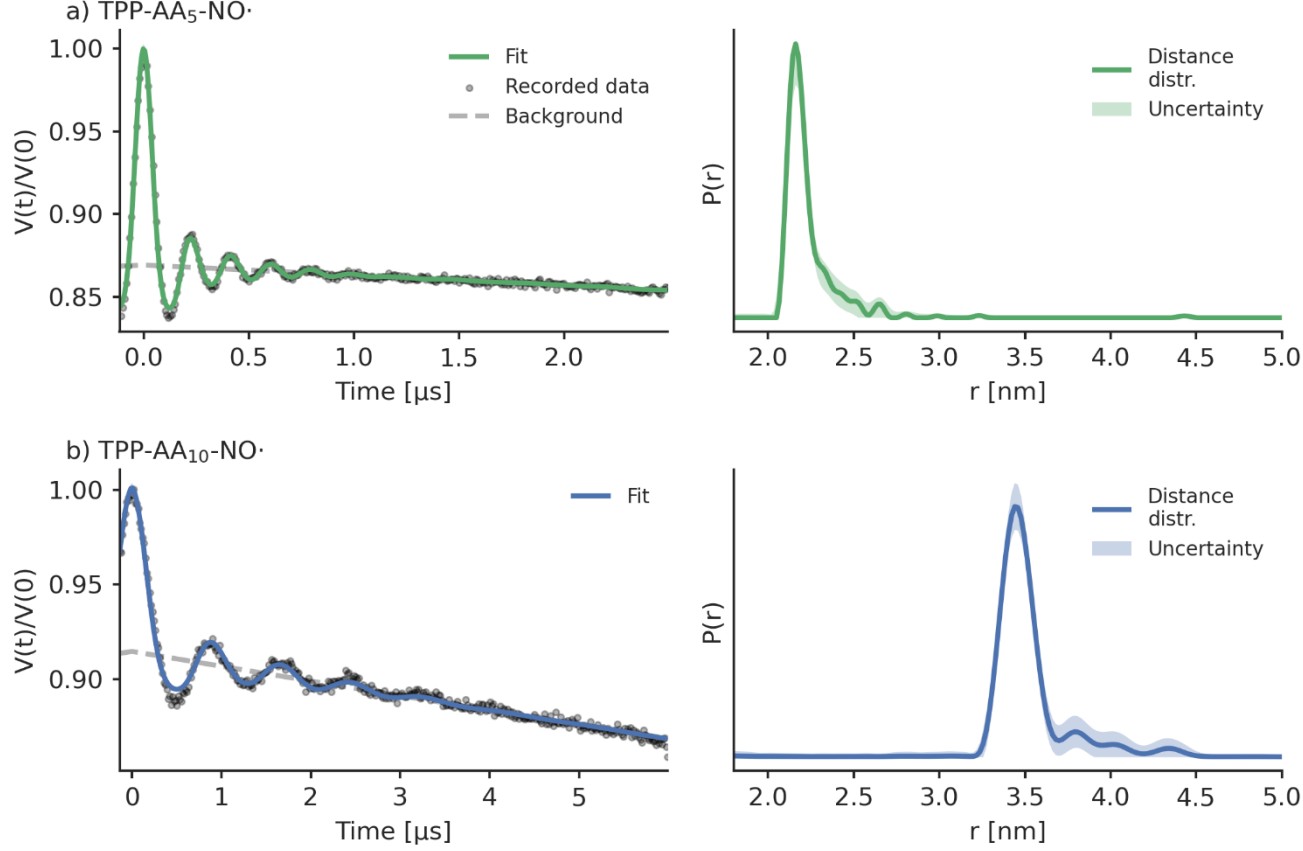

**Figure 11:** Experimental LiDEER data of the two peptides, all recorded in Q-band at 30 K in MeOD/D₂O (98/2 vol.%). a) TPP-pAA₅-NO•
and b) TPP-pAA₁₀-NO•. The raw data are depicted on the left side as grey dots with the fits as straight line, the background fit is depicted
as dashed grey line. The distance distributions obtained with Tikhonov regularization (Fábregas Ibáñez et al., 2020) is shown on the right
side. The shaded areas correspond to the 95% confidence intervals that were obtained with bootstrapping.

So far, the LaserIMD simulations that were described above did mostly only invoke a single delta-like distance. To simulate
LaserIMD for an entire distance distribution in a fast way, the dipolar kernel $K_{\text{LaserIMD}}^{\text{non-sec}}(t,r)$ needs to be calculated. Therefore,
we implemented a C++ software tool that can perform the numerical integration of Eq. (25) -(27) to calculate $S_{\text{LaserIMD}}^{\text{non-sec}}(t,r)$.
It allows the user to specify different ZFS values, zero-field populations and Zeeman frequencies. The background decay and
modulation depth can then be included afterwards to obtain the full kernel $K_{\text{LaserIMD}}^{\text{non-sec}}(t,r)$ (see Eq. (29)). The obtained kernel
can for example be used in combination with the software DeerLab (Fábregas Ibáñez et al., 2020) to analyze experimental
LaserIMD traces. The program including its source-code is available at github (https://github.com/andreas-
scherer/LaserIMD_kernel). Here, it was used to calculate the kernel that corresponds to the experimentally determined
parameters for TPP of the peptides TPP-pAA₅-NO• and TPP-pAA₁₀-NO• (ZFS values: $D = 1159$ MHz and $E = -238$ MHz
and zero-field populations: $P_x = 0.33$, $P_Y = 0.41$ and $P_Z = 0.26$ (Di Valentin et al., 2014)) at the Zeeman frequencies that
correspond to the used magnetic field strengths ($\nu_T = 9.28$ GHz, $\nu_T = 9.31$ GHz in X-band and $\nu_T = 34.00$ GHz in Q-band,

see also S10). The distance distributions of TPP-pAA$_5$-NO• and TPP-pAA$_{10}$-NO• that were used for the LaserIMD simulations were obtained by LiDEER measurements. LiDEER traces were recorded in Q-band with the observer pulse placed on the Y$^-$ peak and analyzed with $K_{\text{DEER}}(t,r)$ and Tikhonov regularization, as the simulations in section 4.2 showed that no artifacts are to be expected in this case. More details on the experiments and distance calculations can be found in S7 and S10. The results

of the LiDEER measurements are shown in Figure 11 and the extracted distance distributions exhibit a narrow peak at 2.2 nm for TPP-pAA$_5$-NO• and at 3.5 nm for TPP-pAA$_{10}$-NO• as expected (Bieber et al., 2018; Di Valentin et al., 2016). As the LaserIMD and LiDEER measurements have different modulation depths, the modulation depth of LiDEER $\lambda_{\text{LiDEER}}$ cannot be used for the simulation of the LaserIMD. This makes the modulation depth of the LaserIMD traces $\lambda_{\text{LaserIMD}}$ the only parameter that is missing for the simulations. Therefore, the simulated LaserIMD traces were fitted to the measured ones by

rescaling the modulation depth. As the background decay rate depends linearly on the modulation depth (Hu and Hartmann, 1974; Pannier et al., 2000), it must be rescaled together with the modulation depth. For LaserIMD, we assume that coherence transfer pathways with $\Delta m_T = 0$ does not contribute to the background, as its decay of the echo intensity is on a much longer timescale than the dipolar oscillations that constitute the main contribution of the intermolecular background. Therefore, we additionally reduce the rescaled background decay rate by a factor of $2/3$:

$$K_{\text{LaserIMD}}^{\text{non-sec}}(t,r)_{\lambda_{\text{LaserIMD}}} = \exp\left(-\frac{2}{3}\frac{\lambda_{\text{LaserIMD}}}{\lambda_{\text{LiDEER}}}k_{\text{LiDEER}}t\right)\left(\lambda_{\text{LaserIMD}}S_{\text{LaserIMD}}^{\text{non-sec}}(t,r) + (1-\lambda_{\text{LaserIMD}})\right) \tag{40}$$

$$V_{\text{LaserIMD}}(t)_{\lambda_{\text{LaserIMD}}} = K_{\text{LaserIMD}}^{\text{non-sec}}(t,r)_{\lambda_{\text{LaserIMD}}}P_{\text{LiDEER}}(r) \tag{41}$$

The simulated LaserIMD trace $V_{\text{LaserIMD}}(t)_{\lambda_{\text{LaserIMD}}}$ was fitted to the experimental LaserIMD data by varying the modulation depth $\lambda_{\text{LaserIMD}}$ so that the root-mean-square displacement of the simulated and experimental traces was minimized. Simulations without the effect of the ZFS were also performed in order to clearly see the difference to the simulations with the ZFS. For the simulations without the ZFS, the modulation depth of the LaserIMD simulations with the ZFS was taken as it was determined by the fit and reduced by a factor of $2/3$, because the coherence transfer pathway with $\Delta m_T = 0$ no longer

contributes to the echo modulation.

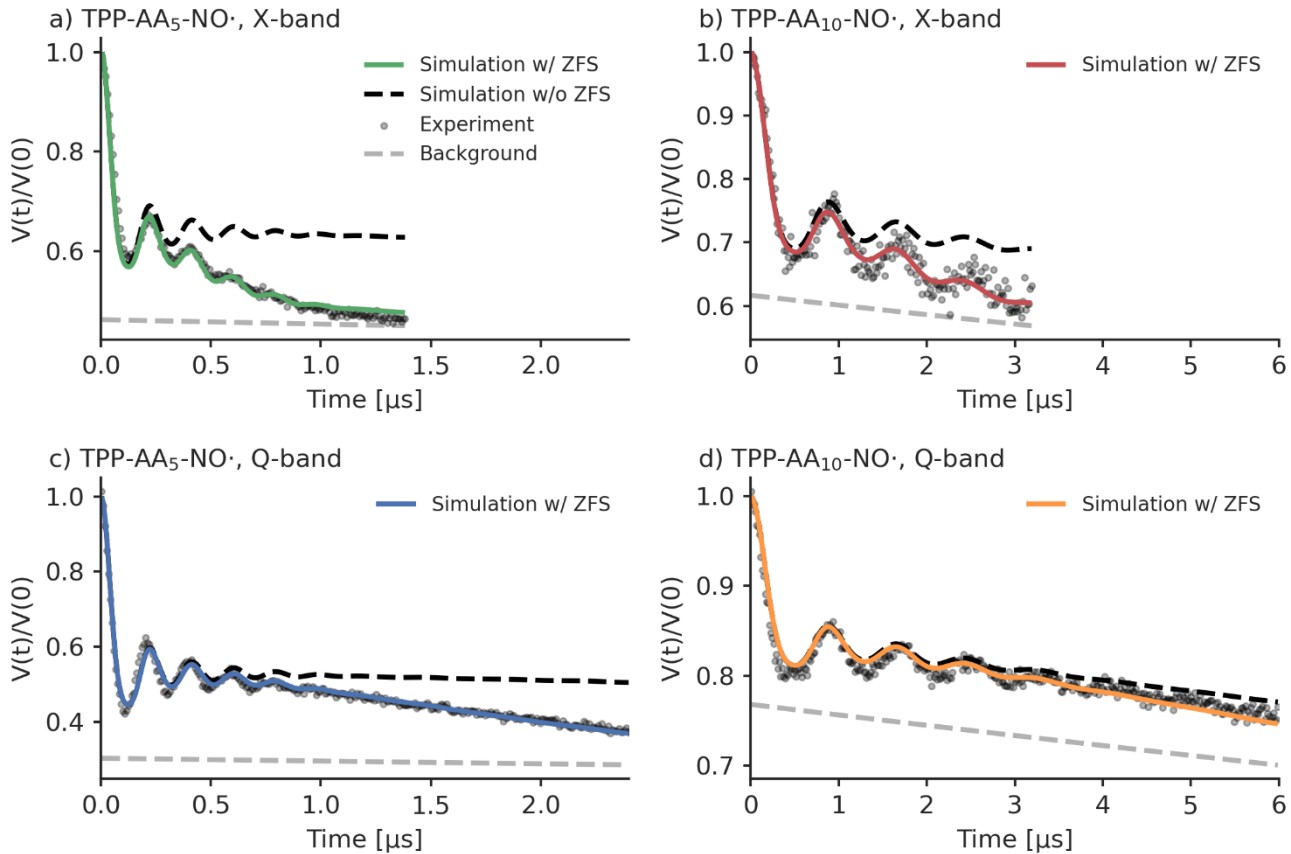

**Figure 12:** Experimental LaserIMD traces of the peptides, recorded at 30 K in MeOD/$D_2O$ (98/2 vol.%). a) TPP-AA$_5$-NO• in X-band ($\nu_T$ = 9.28 GHz) (green), b) TPP-AA$_{10}$-NO• in X-band ($\nu_T$ = 9.31 GHz) (red), c) TPP-AA$_5$-NO• in Q-band ($\nu_T$ = 34.00 GHz) (blue), d) TPP-AA$_{10}$-NO• in Q-band ($\nu_T$ = 34.00 GHz) (orange). The colored traces show simulations that include the ZFS. The simulations without the effects of the ZFS are shown as black dashed line. The experimentally recorded data are depicted as grey dots. The backgrounds of the simulations are shown as grey dashed line. The simulations were performed with the distance distributions and background decays that were obtained by the LiDEER measurements.

The results of the LaserIMD measurements and the corresponding simulations are shown in Figure 12. It can be clearly seen that the shape of the experimental traces changes depending on whether they were recorded in X- or Q-band and with those with a stronger decay in X-band. This is a first strong indication of the effect of the ZFS, as predicted by the simulations (see Figure 5). The influence of the ZFS shows itself clearly in the differences between the experimental data and the simulations where the effect of the ZFS was ignored. In particular, the experimental LaserIMD traces show a stronger decay than the background decay of simulations without the ZFS. This difference is more pronounced in TPP-AA$_5$-NO• than in TPP-AA$_{10}$-NO• and also stronger in X-band than in Q-band. Thus, for TPP-AA$_5$-NO• in X-band, the deviation between the simulations without the ZFS and the experiments is the largest, whereas in the case of TPP-AA$_{10}$-NO• in Q-band, it is nearly absent. This additional decay of the experimental traces cannot be explained without considering the effect of the ZFS, but is properly

understandable with a model that includes the ZFS. The stronger decay of the experimental traces can be assigned to the coherence transfer pathway with $\Delta m_T = 0$, which leads to an additional contribution to the LaserIMD trace $V_0^{\text{non-sec}}(t)$ with a continuously decaying signal (see Figure 4). As shorter distances and lower magnetic fields lead to a stronger decay of $V_0^{\text{non-sec}}(t)$, this also explains why the additional decay in the experimental data is stronger for TPP-AA$_5$-NO• than for TPP-

AA$_{10}$-NO• and stronger in X- than in Q-band. It is noteworthy that the model with the ZFS provides not only a qualitative but also a quantitative agreement between the experimentally recorded LaserIMD traces and the corresponding simulations.

To see how the additional decay of the ZFS affects the analysis of experimental LaserIMD traces, the recorded data were analyzed with Tikhonov regularization; and the results that are obtained with a LaserIMD kernel that includes the ZFS are compared to those obtained by a DEER kernel that ignores the ZFS (see S11 for a detailed overview of the results). The

comparison of the obtained distance distributions shows that, even when the ZFS is ignored, the main distance peak is obtained correctly in all cases. For the measurements in Q-band, the entire distance distributions turn out to be virtually identical, regardless whether the ZFS is included in the analysis routine or not (see Fig. S13c-d). The situation is different in X-band. For TPP-AA$_5$-NO• in X-band, the strong additional decay is interpreted as an additional artifact peak at around 5.0 nm if the ZFS is ignored (see Fig. S13a). This peak disappears when the ZFS is considered. For TPP-AA$_{10}$-NO• in X-band, the analysis

which ignores the ZFS also shows an additional peak around 7.0 nm. However, this artifact is not as pronounced as the one of TPP-AA$_5$-NO• and disappears in the validation. For the modulation depths and the background decay rates, there are notable differences when the ZFS is considered or not (see Table S5 and S6 in S11). In all cases, ignoring the ZFS leads to a reduced modulation depth. In Q-band, the modulation depth is reduced by a factor of $\approx 2/3$ which means that the additional decay is completely assigned to the intermolecular background. In accordance with that, the background decay rates are larger when

the ZFS is ignored. In X-band, these effects are not so pronounced. As the additional decay is partially fitted by introducing distance artifacts when ignoring the ZFS, the modulation depth is reduced only by a factor of 0.72 for TPP-AA$_{10}$-NO• and 0.84 for TPP-AA$_5$-NO•.

These results show that ignoring the ZFS for the analysis of LaserIMD leads to artifacts in the obtained results. For TPP as transient spin label, the artifacts are not so prominent in Q-band. There, the additional decay mostly leads to a stronger

background decay and reduced modulation depth and the distance distribution remains virtually unchanged. In X-band, however, artifact peaks in the distance distribution can occur if the ZFS is ignored.

## 5 Conclusion and Outlook

In light-induced PDS, the ZFS interaction of the transient triplet label is a crucial parameter that can alter the shape of the dipolar traces. This implies that in contrast to the former assumption, the spin system in LaserIMD and LiDEER cannot be

treated in the secular approximation where the spin system behaves as if it would consist of two $S = 1/2$ spins. A theoretical description of LaserIMD and LiDEER that also includes non-secular terms was developed and it was shown that the dipolar frequencies depend on the magnitude of the ZFS and the Zeeman frequency (i.e. the external magnetic field). Time-domain

simulations showed that in LiDEER, this effects of the ZFS can be suppressed by exciting either of the $Y^{\pm}$ peaks with the observer pulses and by using transient triplet labels whose ZFS is small compared to the Zeeman frequency, like e.g. TPP in Q-band. For experimental LiDEER data which are recorded under such conditions the effect of the ZFS is negligible and a standard DEER kernel that does not consider the ZFS can be employed for data analysis.

In LaserIMD, simulations as well as experiments confirmed that there is an influence of the ZFS on the dipolar trace. It virtually does not affect the dipolar oscillation of the coherence transfer pathways with $\Delta m_{\mathrm{T}} = \pm 1$, but is manifested in an additional decay of the LaserIMD trace. This decay is caused by the third coherence transfer pathway with $\Delta m_{\mathrm{T}} = 0$, which was formerly believed not to contribute to the signal. The strength of this additional decay primarily depends on the ratio of the ZFS to the Zeeman frequency and also the distance between the transient and permanent spin label: It is stronger for larger ZFS, lower

magnetic fields and shorter distances. A software tool for the calculation of LaserIMD kernels that take the influence of the ZFS into account was developed. It is available at github (https://github.com/andreas-scherer/LaserIMD_kernel) and allows to specify different ZFS values, zero-field populations and Zeeman frequencies. The feasibility of the new kernel was proven by experimentally recorded LaserIMD traces. A DEER kernel which ignores the ZFS cannot fit these traces correctly and strong derivations between the experimental data and simulations can be observed. However, with the newly developed model that

considers the ZFS, excellent fits of the experimental data were produced. The analysis of the experimental and simulated LaserIMD data with Tikhonov regularization showed that ignoring the ZFS compromises the obtained results. For transient triplet labels with a ZFS of $\approx 1$ GHz like TPP, this is no so problematic in Q-band. There, only the obtained modulation depths and background decay rates are affected if the ZFS is ignored; the distance distribution remains unchanged. In X-band, however, ignoring the ZFS is more severe and can additionally lead to artifact peaks in the distance distributions. This shows

that the ZFS can have a significant impact in LaserIMD and should be considered when experimental data are analyzed.

**6 Data availability**

The raw data can be downloaded at https://doi.org/10.5281/zenodo.7283499.

**7 Code availability**

The source code for the LaserIMD kernel can be downloaded at https://github.com/andreas-scherer/LaserIMD_kernel. The
source code for the time-domain LiDEER simulations can be downloaded at https://github.com/andreas-scherer/LiDEER_simulations.git

**8 Author contribution**

AS and MD conceived the research idea and designed the simulations and experiments. AS performed the analytical calculations and AS and BY conducted the simulations and experiments and analyzed the results. AS prepared all the figures and wrote the draft manuscript. All authors discussed the results and revised the manuscript.

**9 Competing interests**

The authors declare no conflict of interests.

**10 Acknowledgements**

We thank Joschua Braun and Stefan Volkwein for helpful discussions concerning the numerical integration for the LaserIMD kernel. This project has received funding from the European Research Council (ERC) under the European Union's Horizon 2020 research and innovation programme (Grant Agreement number: 772027 — SPICE — ERC-2017-COG). AS gratefully acknowledge financial support from the Konstanz Research School Chemical Biology (KoRS-CB).

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
