# Peer review of "The effect of the zero-field splitting in light-induced pulsed dipolar EPR spectroscopy"

_Magnetic Resonance, 2022_

## Author Comment (AC3)

Our answers are written in green, changes in the manuscript are in blue.

1. In the abstract and conclusions of the current version of the manuscript, the authors are quite vague on the extent to which "the ZFS cannot be neglected" in the analysis of dipolar traces. It appears that in all cases that could reasonably be expected to be encountered, accurate distance distributions are obtained even with the standard S=1/2 kernel (except for clear artifacts at the end of the distance range accessible for the given length of the trace). This should be made clearer in the abstract and conclusions.

**The mentioned sentence of the abstract was replaced with the following sentence: Here, we explore the limits of this assumption and show that the ZFS can have a significant effect on the shape of the dipolar trace.**

The last sentence of the abstract was replaced by the following paragraph:

Experimentally recorded LiDEER and LaserIMD data confirm these findings. It is shown that ignoring the ZFS in the data analysis of LaserIMD traces can lead to errors in the obtained modulation depths and background decays. In X-band, it is additionally possible to that the obtained distance distribution is plagued by long distance artifacts.

The following paragraph was added at the end of the conclusion:

The analysis of the experimental and simulated LaserIMD data with Tikhonov regularization showed that ignoring the ZFS compromises the obtained results. For transient triplet labels with a ZFS of  $\approx$ 1 GHz like TPP, this is no so problematic in Q-band. There, only the obtained modulation depths and background decay rates are affected if the ZFS is ignored and the distance distribution remains unchanged. In X-band, however, ignoring the ZFS is more severe and can additionally lead to artifact peaks in the distance distributions.

2. In the initial discussion of the different triplet labels that have been shown to be appropriate for light-induced PDS it would be useful to provide the corresponding D and E values, given the later discussion of spin labels with small or large ZFS D parameters and the corresponding effects on the dipolar traces. Since this manuscript was first made available online, an additional paper discussing the use of erythrosin B as a triplet spin label has been published and should be referenced as well (DOI: 10.3390/molecules27217526).

Starting form p. line 4 of the original manuscript, the paragraph was rephrased:

In contrast to spin labels with a spin of S = 1/2 like nitroxides, these transient triplet labels are subject to an additional zero-field splitting (ZFS). It is described by the ZFS parameters D and E. By now, several transient triplet labels with different ZFS strengths have been used. Examples are triphenylporphyrin (TPP) (D = 1159 MHz, E = -238 MHz) (Di Valentin et al., 2014), fullerenes (D = 342 MHz, E = -2 MHz) (Wasielewski et al., 1991; Krumkacheva et al., 2019; Timofeev et al., 2022), Rose Bengal (D =3671 MHz, E = -319 MHz), Eosin Y (D = 2054 MHz, E = -585 MHz), Atto Thio12 (D =1638 MHz, E = -375 MHz) (Serrer et al., 2019; Williams et al., 2020) and Erythrosin B (D =3486 MHz, E = -328 MHz) (Bertran et al., 2022b).

3. The authors should clarify in the main manuscript which non-secular terms of the Hamiltonian are included in their treatment and based on what arguments. The current statement "we did not consider all non-secular terms and pseudo-secular terms were also ignored" at the end of page 7 (and in the SI) is insufficient and not clear. I would also recommend clarifying in the main manuscript that non-secular terms are considered both for the zero-field interaction (S\_T\*D\*S\_T) and for the dipole-dipole interaction (S\_T\*T\*S\_D), since previous treatments of systems with ZFS have focused on the effect of the pseudo-secular term of the dipole-dipole interaction on PDS traces (DOI: 10.1063/1.4994084).

We rephrased the corresponding paragraph starting from p.7, l. 12 of the original manuscript:

Even though in the secular approximation the ZFS has no effect in LaserIMD, it cannot be taken for granted that the non-secular terms can be ignored because the ZFS of some transient triplet labels can be quite large (Williams et al., 2020). Here, we additionally consider the terms  $\hat{S}_{T,z}\hat{S}_{T,+} + \hat{S}_{T,+}\hat{S}_{T,z}$  and  $\hat{S}_{T,-}\hat{S}_{T,z} + \hat{S}_{T,-}\hat{S}_{T,z}$  from the ZFS interaction and the terms  $\hat{S}_{D,z}\hat{S}_{T,+}$  and  $\hat{S}_{D,z}\hat{S}_{T,-}$  from the dipolar coupling. They connect the adjacent triplet states  $|+1\rangle$  and  $|0\rangle$  and  $|0\rangle$  and  $|-1\rangle$  of the triplet manifold and shift their energy in second order (Hagston and Holmes, 1980). This is illustrated in Fehler! Verweisquelle konnte nicht gefunden werden.. The details of this calculation are described in S1. For this calculation, the

remaining ZFS terms  $\hat{S}_{T,+}^2$  and  $\hat{S}_{T,-}^2$  were ignored. They connect the triplet states  $| + 1 \rangle$  and  $| - 1 \rangle$ , which have a larger energy difference than adjacent states. Therefore, the second order energy shift of  $\hat{S}_{T,+}^2$  and  $\hat{S}_{T,-}^2$  is weaker than those of the considered terms. The terms  $\hat{S}_{D,+}\hat{S}_{T,+}$ ,  $\hat{S}_{D,-}\hat{S}_{T,+}$ ,  $\hat{S}_{D,+}\hat{S}_{T,-}$ ,  $\hat{S}_{D,-}\hat{S}_{T,-}$ ,  $\hat{S}_{D,+}\hat{S}_{T,z}$  and  $\hat{S}_{D,-}\hat{S}_{T,z}$  of the dipolar coupling were also ignored. They connect spin states of different manifolds of the doublet spin and the corresponding energies cannot be significantly shifted by the comparably weak dipolar coupling. It is shown in S2 that at magnetic field strengths that are relevant for experimental conditions the included non-secular terms from Eq. (S3) are sufficient and no further distortions are to be expected by the left-out ones.

We also deleted these sentences from S1, p.3, l.4-7 to avoid repetition:

Here,  $\hat{S}_{T,+}$  and  $\hat{S}_{D,+}$  are the raising  $(\hat{S}_{+} = \hat{S}_{x} + i\hat{S}_{y})$  and  $\hat{S}_{D,-}$  and  $\hat{S}_{T,-}$  are the lowering  $(\hat{S}_{-} = \hat{S}_{x} - i\hat{S}_{y})$  operators. Note that this non-secular part does not include all non-secular terms and all-pseudo secular terms are also ignored here. As is shown in S2 these remaining terms have only a negligible effect at the magnetic fields strengths that are relevant in most experiments, and this is why they can be left out.

4. All of the simulations aimed at illustrating the effects of different spin system and experimental parameters on the dipolar trace have been performed for a single distance and show modulations that persist for a very long time. All of the systems encountered experimentally will be characterized by a distribution of distances. I believe it would be more instructive and more meaningful to show simulations performed for distance distributions in addition to or instead of the single-distance simulations as these would be closer to traces encountered experimentally and allow users to judge what effect they could expect in experimentally relevant situations. For example, it would be interesting to see how the V\_mS=0 contribution would be affected by distance distributions with increasing widths. Also, would the effects illustrated in Figures 6, 8 and S8 even be visible in the presence of a distance distribution rather than a single distance?

We performed additional simulations with an entire distance distribution with different widths. We added the following paragraph starting from p.15 l.14 to:

[revised manuscript text omitted]

6. It would be very useful to compare distance distributions extracted from the experimental LaserIMD data using the standard S=1/2 kernel and the correct kernel including the effects of the non-secular terms in the Hamiltonian in Figure 10. It appears that the effect of the V\_mS=0 contribution would mostly be compensated by the background contribution if the S=1/2 kernel is used and therefore not significantly affect the extracted distance distribution. Therefore, if only the distance distribution is of interest, it would appear that the standard analysis should be sufficient in almost all cases.

We performed the suggested analysis and added to results in the SI. They are discussed in the following paragraph which is added at the end of manuscript:

To see how the additional decay of the ZFS affects the analysis of experimental LaserIMD traces, the recorded data were analyzed with Tikhonov regularization and the results that are obtained with a LaserIMD kernel that includes the ZFS and a DEER kernel that ignores the ZFS are compared (see S11 for a detailed overview of the results). The comparison of the obtained distance distributions shows that, even when the ZFS is ignored, the main distance peak is obtained correctly in all cases. For the measurements in Q-band, the entire distance distributions turn out to be virtually identical, regardless whether the ZFS is included in the analysis routine or not (see Fig. S13c-d). The situation is different in X-band. For TPP-AA5-NO• in X-band, the strong additional decay is interpreted as an additional artifact peak at around 5.0 nm if the ZFS is ignored (see Fig. S13a). This peak disappears when the ZFS is considered. For TPP-AA10-NO• in X-band, the analysis which ignores the ZFS also shows an additional peak around 7.0 nm. However, this artifacts peak is not as pronounced as the one of TPP AA5 NO• and disappears in the validation. For the modulation depths and the background decay rates, there are notable differences when the ZFS is considered or not (see Table S5 and S6 in S11). In all cases, ignoring the ZFS leads to a reduced modulation depth. In Q-band, the modulation depth is reduced by a factor of  $\approx 2/3$  which means that the additional

decay is completely assigned to the intermolecular background. In accordance with that, the background decay rates are larger when the ZFS is ignored. In X-band, these effects are not so pronounced. As the additional decay is partially fitted by introducing distance artifacts when ignoring the ZFS, the modulation depth is reduced only by a factor of 0.72 for TPP AA10-NO• and 0.84 for TPP-AA5-NO•.

These results show that ignoring the ZFS for the analysis of LaserIMD leads to artifacts in the obtained results. For TPP as transient spin label, the artifacts are not so prominent in Q-band. There, the additional decay mostly leads to a stronger background decay and reduced modulation depth and the distance distribution remains virtually unchanged. In X-band, however, artifact peaks in the distance distribution can occur if the ZFS is ignored.

The following paragraph was added in the conclusion:

The analysis of the experimental and simulated LaserIMD data with Tikhonov regularization showed that ignoring the ZFS compromises the obtained results. For transient triplet labels with a ZFS of  $\approx$ 1 GHz like TPP, this is no so problematic in Q-band. There, only the obtained modulation depths and background decay rates are affected by ignoring the ZFS and the distance distribution remains unchanged. In X-band, however, ignoring the ZFS is more severe and can additionally lead to artifact peaks in the distance distributions.

A new section S11 was added in the SI:

**S11 Analysis of the experimental LaserIMD data**

**Figure S1:** Experimental LaserIMD data of TPP-pAA5-NO• recorded in X-band at 30 K in MeOD/D2O (98/2 vol.%). a) Analyzed with a kernel that includes the ZFS and b) Analyzed with a kernel that ignores the ZFS. The raw data are depicted on the left side as grey dots with the fits as green line, the background fit is depicted as dashed grey line. The distance distributions obtained with Tikhonov regularization (Fábregas Ibáñez et al., 2020) is shown on the right side. The shaded areas correspond to the 95% confidence intervals that were obtained with bootstrapping.

---

## Author Response (AR1)

**Reviewer 1:**

Our answers are written in green, changes in the manuscript are in blue.

1. As I understand it, all analysis in terms of distance distributions in this work has been performed with the DEER kernel (S = 1/2 approximation), while you argue that the kernel should include effects due to the ZFS and provide software for computing such adapted kernels. You analyze in terms of distance distributions with the open-source software DeerLab. Why don't you directly compare analysis by Tikhonov regularization with the DEER kernel and your kernel? This would be particularly valuable for your experimental data.

We performed the suggested analysis and added to results in the SI. They are discussed in the following paragraph which is added to the manuscript:

To see how the additional decay of the ZFS affects the analysis of experimental LaserIMD traces, the recorded data were analyzed with Tikhonov regularization; and the results that are obtained with a LaserIMD kernel that includes the ZFS are compared with those obtained by a DEER kernel that ignores the ZFS (see S11 for a detailed overview of the results). The comparison of the obtained distance distributions shows that, even when the ZFS is ignored, the main distance peak is obtained correctly in all cases. For the measurements in Q-band, the entire distance distributions turn out to be virtually identical, regardless whether the ZFS is included in the analysis routine or not (see Fig. S13c-d). The situation is different in X-band. For TPP-AA$_5$-NO• in X-band, the strong additional decay is interpreted as an additional artifact peak at around 5.0 nm if the ZFS is ignored (see Fig. S13a). This peak disappears when the ZFS is considered. For TPP-AA$_{10}$-NO• in X-band, the analysis which ignores the ZFS also shows an additional peak around 7.0 nm. However, this artifact is not as pronounced as the one of TPP AA5 NO• and disappears in the validation. For the modulation depths and the background decay rates, there are notable differences when the ZFS is considered or not (see Table S5 and S6 in S11). In all cases, ignoring the ZFS leads to a reduced modulation depth. In Q-band, the modulation depth is reduced by a factor of ≈2/3 which means that the additional decay is completely assigned to the intermolecular background. In accordance with that, the background decay rates are larger when the ZFS is ignored. In X-band, these effects are not so pronounced. As the additional decay is partially fitted by introducing distance artifacts when ignoring the ZFS, the modulation depth is reduced only by a factor of 0.72 for TPP AA10-NO• and 0.84 for TPP-AA$_5$-NO•.
These results show that ignoring the ZFS for the analysis of LaserIMD leads to artifacts in the obtained results. For TPP as transient spin label, the artifacts are not so prominent in Q-band. There, the additional decay mostly leads to a stronger background decay and reduced modulation depth and the distance distribution remains virtually unchanged. In X-band, however, artifact peaks in the distance distribution can occur if the ZFS is ignored.

The following paragraph was added in the conclusion:

The analysis of the experimental and simulated LaserIMD data with Tikhonov regularization showed that ignoring the ZFS compromises the obtained results. For transient triplet labels with a ZFS of ≈1 GHz like TPP, this is no so problematic in Q-band. There, only the obtained modulation depths and background decay rates are affected by ignoring the ZFS and the distance distribution remains unchanged. In X-band, however, ignoring the ZFS is more severe and can additionally lead to artifact peaks in the distance distributions.

A new section S11 was added in the SI:

**S11 Analysis of the experimental LaserIMD data**

[Figure]

**Figure S1:** Experimental LaserIMD data of TPP-pAA$_5$-NO• recorded in X-band at 30 K in MeOD/D$_2$O (98/2 vol.%). a) Analyzed with a kernel that includes the ZFS and b) Analyzed with a kernel that ignores the ZFS. The raw data are depicted on the left side as grey dots with the fits as green line, the background fit is depicted as dashed grey line. The distance distributions obtained with Tikhonov regularization (Fábregas Ibáñez et al., 2020) is shown on the right side. The shaded areas correspond to the 95% confidence intervals that were obtained with bootstrapping.

[Figure]

**Figure S2:** Experimental LaserIMD data of TPP-pAA$_5$-NO• recorded in Q-band at 30 K in MeOD/D$_2$O (98/2 vol.%). a) Analyzed with a kernel that includes the ZFS and b) Analyzed with a kernel that ignores the ZFS. The raw data are depicted on the left side as grey dots with the fits as blue line, the background fit is depicted as dashed grey line. The distance distributions obtained with Tikhonov regularization (Fábregas Ibáñez et al., 2020) is shown on the right side. The shaded areas correspond to the 95% confidence intervals that were obtained with bootstrapping.

[Figure]

**Figure S3:** Experimental LaserIMD data of TPP-pAA$_{10}$-NO• recorded in X-band at 30 K in MeOD/D$_2$O (98/2 vol.%). a) Analyzed with a kernel that includes the ZFS and b) Analyzed with a kernel that ignores the ZFS. The raw data are depicted on the left side as grey dots with the fits as red line, the background fit is depicted as dashed grey line. The distance distributions obtained with Tikhonov regularization (Fábregas Ibáñez et al., 2020) is shown on the right side. The shaded areas correspond to the 95% confidence intervals that were obtained with bootstrapping.

[Figure]

**Figure S4:** Experimental LaserIMD data of TPP-pAA$_{10}$-NO• recorded in Q-band at 30 K in MeOD/D$_2$O (98/2 vol.%). a) Analyzed with a kernel that includes the ZFS and b) Analyzed with a kernel that ignores the ZFS. The raw data are depicted on the left side as grey dots with the fits as orange line, the background fit is depicted as dashed grey line. The distance distributions obtained with Tikhonov regularization **(Fábregas Ibáñez et al., 2020)** is shown on the right side. The shaded areas correspond to the 95% confidence intervals that were obtained with bootstrapping.

[Figure]

**Figure S5:** A comparison of the distance distributions that were obtained by analyzing the experimental LaserIMD data with a kernel that includes the ZFS (coloured lines) and with a kernel that ignores the FS (black lines). a) TPP-pAA$_5$-NO• in X-band. b) TPP-pAA$_5$-NO• in Q band. c) TPP-pAA$_{10}$-NO• in X-band. a) TPP-pAA$_{10}$-NO• in Q-band.

**Table S1: Background decay rates and modulation depths as obtained by the analysis of the LaserIMD data of TPP-pAA$_5$-NO•.**

| | X-band | | Q-band | |
|---|---|---|---|---|
| | w/ ZFS | w/o ZFS | w/ ZFS | w/o ZFS |
| **Background decay rate [$\mu s^{-1}$]** | 0.0 (0.0, 0.2) | 0.1 (0.0, 0.3) | 0.000 (0.003, 0.000) | 0.33 (0.27, 0.36) |
| **Modulation depth [%]** | 57 (52, 59) | 48 (43, 52) | 71 (70, 72) | 46 (45, 49) |

**Table S6: Background decay rates and modulation depths as obtained by the analysis of the LaserIMD data of TPP-pAA$_{10}$-NO•.**

| | X-band | | Q-band | |
|---|---|---|---|---|
| | w/ ZFS | w/o ZFS | w/ ZFS | w/o ZFS |
| **Background decay rate [$\mu s^{-1}$]** | 0.00 (0.00, 0.05) | 0.13 (0.01, 0.28) | 0.03 (0.01, 0.04) | 0.09 (0.07, 0.11) |
| **Modulation depth [%]** | 43 (40, 45) | 31 (25, 36) | 26 (25, 27) | 17 (16, 18) |

2. Except perhaps for the case of LiDEER performed at non-canonical orientations in X band, effects of ZFS on the extracted distance distribution are so minor that they are likely overwhelmed by other uncertainties in application work. If you agree with this assessment, you should clearly state this in the Conclusion.

The following sentence was added on p. 26, l. 15:
"For experimental LiDEER data which are recorded under such conditions the effect of the ZFS is negligible and a standard DEER kernel that does not consider the ZFS can be employed for data analysis."

3. I think that the experimental data is underused. Even if you perform only simulations with your own kernel (instead of using it in Tikhonov regularization), you should make an effort to assess the influence that ZFS has on the background decay rate and modulation depth for these examples.

We think that this is also covered by the answer to point one.

4. Your referencing does not follow established rules. If you provide a reference for a statement, it should be either the first paper where this was found or a review/book chapter. If the statement can be considered as textbook knowledge, no reference is needed. In several cases you rather appear to cite the papers where you first encountered the same statement. For example, you cite me for textbook knowledge (distance dependence of the dipolar coupling) and for work by Salikhov, Tsvetkov, and Milov (p. 4, l. 11, citation (Jeschke, 2016) for the term "background", if this really needs a citation). There are many more instances, also affecting others. In a very general Introduction as you write it, the absence of citations to the pioneering work from the Novosibirsk lab is problematic.

We updated the referencing to fit common usage. The references to textbook knowledge were deleted and other citations were corrected to the original work. We also added the work from Novosibirsk.

5. In the Introduction, you come close to considering orientation selection, but you never mention it. You should do so, as neglect of orientation selection is a feature of your treatment.

On page 6, line 9 the following sentence was added:
"In absence of orientation selection, the orientation of the dipolar vector and the transient triplet label are not correlated and the integration over the corresponding Euler angles can be done independently."

On page 9, line 9, the following sentence was added:
"Still assuming no orientation selection, this gives the following integrals."

6. "Please note that we did not consider all non-secular terms and pseudo-secular terms were also ignored." It is not clear to me, which terms you consider as pseudo-secular and how you selected the terms that you included. Section S1 of the Supplementary Information does not help. Common usage is that terms that you consider on top of the secular terms are pseudo-secular and terms that you drop are (considered as) non-secular.

We rephrased the corresponding paragraph starting from p.7, l. 12:
Even though in the secular approximation the ZFS has no effect in LaserIMD, it cannot be taken for granted that the non-secular terms can be ignored because the ZFS of some transient triplet labels can be quite large (Williams et al., 2020). Here, we additionally consider the terms $\widehat{S}_{T,z}\widehat{S}_{T,+} + \widehat{S}_{T,+}\widehat{S}_{T,z}$ and $\widehat{S}_{T,-}\widehat{S}_{T,z} + \widehat{S}_{T,-}\widehat{S}_{T,z}$ from the ZFS interaction and the terms $\widehat{S}_{D,z}\widehat{S}_{T,+}$ and $\widehat{S}_{D,z}\widehat{S}_{T,-}$ from the dipolar coupling. They connect the adjacent triplet states $|+1\rangle$ and $|0\rangle$ and $|0\rangle$ and $|-1\rangle$ of the triplet manifold and shift their energy in second order (Hagston and Holmes, 1980). This is illustrated in Fig. 2. The details of this calculation are described in S1. For this calculation, the remaining ZFS terms $\widehat{S}_{T,+}^2$ and $\widehat{S}_{T,-}^2$ were ignored. They connect the triplet states $|+1\rangle$ and $|-1\rangle$, which have a larger energy difference than adjacent states. Therefore, the second order energy shift of $\widehat{S}_{T,+}^2$ and $\widehat{S}_{T,-}^2$ is weaker than those of the considered terms. The terms $\widehat{S}_{D,+}\widehat{S}_{T,+}$, $\widehat{S}_{D,-}\widehat{S}_{T,+}$, $\widehat{S}_{D,+}\widehat{S}_{T,-}$, $\widehat{S}_{D,-}\widehat{S}_{T,-}$, $\widehat{S}_{D,+}\widehat{S}_{T,z}$ and $\widehat{S}_{D,-}\widehat{S}_{T,z}$ of the dipolar coupling were also ignored. They connect spin states of different manifolds of the doublet spin and the corresponding energies cannot be significantly shifted by the comparably weak dipolar coupling. It is shown in S2 that at magnetic field strengths that are relevant for experimental conditions the included non-secular terms from Eq. (S3) are sufficient and no further distortions are to be expected by the left-out ones.

The following sentences starting from p.7, l.13 were deleted:

~~In Fig. 2 it is shown, how the energy levels E_(m_D, m_T)^(non-sec) get shifted when additional non-secular terms of the ZFS and dipolar coupling are considered. This shift was calculated by including the non-secular terms from Eq. (S3) in S1 and diagonalizing the Hamiltonian with a second-order perturbation approach (Hagston and Holmes, 1980). This is described in detail in S1. Please note that we did not consider all non-secular terms and pseudo-secular terms were also ignored.~~

We also deleted these sentences from S1, p.3, l.4-7 to avoid repetition:

~~Here, $\hat{S}_{T,+}$ and $\hat{S}_{D,+}$ are the raising ($\hat{S}_+ = \hat{S}_x + i\hat{S}_y$) and $\hat{S}_{D,-}$ and $\hat{S}_{T,-}$ are the lowering ($\hat{S}_- = \hat{S}_x - i\hat{S}_y$) operators. Note that this non-secular part does not include all non-secular terms and all pseudo-secular terms are also ignored here. As is shown in S2 these remaining terms have only a negligible effect at the magnetic fields strengths that are relevant in most experiments, and this is why they can be left out.~~

7. In powder averaging, an equidistant grid over cos $\beta_{dip}$ would have been more efficient (all grid points would have had the same weight). I do not suggest that you repeat the work. This is just advice for future work.

   Thank you very much for the advice.

8. p. 5, l. 10: "The dipolar coupling tensor is axial". This presumes the point-dipole approximation, which might be questionable for a TPP triplet at the shorter distance of 2.2 nm. In any case you should mention that your treatment uses the point-dipole approximation.

   We added a half sentence to clarify this point:
   "In the point-dipole approximation, the dipolar coupling tensor $T$ is axial with the eigenvalues $T_x = T_y = -\omega_{dip}$ and $T_z = 2\omega_{dip}$."

9. 8. p. 12, l. 13: "In X-band the resonator was critically coupled to a Q-value of ≈ 900-2000 for higher sensitivity". Did you check this? A higher Q improves detection sensitivity, but reduces excitation bandwidth. Common wisdom is that, as long as you have sufficient microwave power, you should overcouple. What is different in your case?

   Bieber et al. found that LaserIMD performs better in critically coupled than in overcoupled resonators (Bieber et al., 2018). However, this was done on a different spectrometer in Q-band and we did not check it again in X-band. Therefore, we deleted the last part:
   "In X-band the resonator was critically coupled to a Q-value of ≈ 900-2000 ."

10. p. 13, l. 8: "effects of the background were ignored": You probably want to say that background decay was ignored.

    This was changed to:
    "no background was added"

11. Typos

    The typos were corrected.

**Reviewer 2**

Our answers are written in green, changes in the manuscript are in blue.

12. In the abstract and conclusions of the current version of the manuscript, the authors are quite vague on the extent to which "the ZFS cannot be neglected" in the analysis of dipolar traces. It appears that in all cases that could reasonably be expected to be encountered, accurate distance distributions are obtained even with the standard S=1/2 kernel (except for clear artifacts at the end of the distance range accessible for the given length of the trace). This should be made clearer in the abstract and conclusions.

The mentioned sentence of the abstract was replaced with the following sentence:
Here, we explore the limits of this assumption and show that the ZFS can have a significant effect on the shape of the dipolar trace.

The last sentence of the abstract was replaced by the following paragraph:
Experimentally recorded LiDEER and LaserIMD data confirm these findings. It is shown that ignoring the ZFS in the data analysis of LaserIMD traces can lead to errors in the obtained modulation depths and background decays. In X-band, it is additionally possible to that the obtained distance distribution is plagued by long distance artifacts.

The following paragraph was added at the end of the conclusion:
The analysis of the experimental and simulated LaserIMD data with Tikhonov regularization showed that ignoring the ZFS compromises the obtained results. For transient triplet labels with a ZFS of ≈1 GHz like TPP, this is no so problematic in Q-band. There, only the obtained modulation depths and background decay rates are affected if the ZFS is ignored and the distance distribution remains unchanged. In X-band, however, ignoring the ZFS is more severe and can additionally lead to artifact peaks in the distance distributions.

13. In the initial discussion of the different triplet labels that have been shown to be appropriate for light-induced PDS it would be useful to provide the corresponding D and E values, given the later discussion of spin labels with small or large ZFS D parameters and the corresponding effects on the dipolar traces. Since this manuscript was first made available online, an additional paper discussing the use of erythrosin B as a triplet spin label has been published and should be referenced as well (DOI: 10.3390/molecules27217526).

Starting form p. line 4 of the original manuscript, the paragraph was rephrased:
In contrast to spin labels with a spin of $S = 1/2$ like nitroxides, these transient triplet labels are subject to an additional zero-field splitting (ZFS). It is described by the ZFS parameters $D$ and $E$. By now, several transient triplet labels with different ZFS strengths have been used. Examples are triphenylporphyrin (TPP) ($D = 1159 \, \text{MHz}, E = -238 \, \text{MHz}$ ) (Di Valentin et al., 2014), fullerenes ($D = 342 \, \text{MHz}, E = -2 \, \text{MHz}$ ) (Wasielewski et al., 1991; Krumkacheva et al., 2019; Timofeev et al., 2022), Rose Bengal ($D = 3671 \, \text{MHz}, E = -319 \, \text{MHz}$ ), Eosin Y ($D = 2054 \, \text{MHz}, E = -585 \, \text{MHz}$ ), Atto Thio12 ($D = 1638 \, \text{MHz}, E = -375 \, \text{MHz}$ ) (Serrer et al., 2019; Williams et al., 2020) and Erythrosin B ($D = 3486 \, \text{MHz}, E = -328 \, \text{MHz}$ ) (Bertran et al., 2022b).

14. The authors should clarify in the main manuscript which non-secular terms of the Hamiltonian are included in their treatment and based on what arguments. The current statement "we did not consider all non-secular terms and pseudo-secular terms were also ignored" at the end of page 7 (and in the SI) is insufficient and not clear. I would also recommend clarifying in the main manuscript that non-secular terms are considered both for the zero-field interaction (S_T*D*S_T) and for the dipole-dipole interaction (S_T*T*S_D), since previous treatments of systems with ZFS have focused on the effect of the pseudo-secular term of the dipole-dipole interaction on PDS traces (DOI: 10.1063/1.4994084).

We rephrased the corresponding paragraph starting from p.7, l. 12 of the original manuscript:
Even though in the secular approximation the ZFS has no effect in LaserIMD, it cannot be taken for granted that the non-secular terms can be ignored because the ZFS of some transient triplet labels can be quite large (Williams et al., 2020). Here, we additionally consider the terms $\widehat{S}_{T,z}\widehat{S}_{T,+} + \widehat{S}_{T,+}\widehat{S}_{T,z}$ and $\widehat{S}_{T,-}\widehat{S}_{T,z} + \widehat{S}_{T,-}\widehat{S}_{T,z}$ from the ZFS interaction and the terms $\widehat{S}_{D,z}\widehat{S}_{T,+}$ and $\widehat{S}_{D,z}\widehat{S}_{T,-}$ from the dipolar coupling. They connect the adjacent triplet states $|+1\rangle$ and $|0\rangle$ and $|0\rangle$ and $|-1\rangle$ of the triplet manifold and shift their energy in second order (Hagston and Holmes, 1980). This is illustrated in Fehler! Verweisquelle konnte

nicht gefunden werden.. The details of this calculation are described in S1. For this calculation, the remaining ZFS terms $\widehat{S}_{T,+}^2$ and $\widehat{S}_{T,-}^2$ were ignored. They connect the triplet states $|+1\rangle$ and $|-1\rangle$, which have a larger energy difference than adjacent states. Therefore, the second order energy shift of $\widehat{S}_{T,+}^2$ and $\widehat{S}_{T,-}^2$ is weaker than those of the considered terms. The terms $\widehat{S}_{D,+}\widehat{S}_{T,+}$, $\widehat{S}_{D,-}\widehat{S}_{T,+}$, $\widehat{S}_{D,+}\widehat{S}_{T,-}$, $\widehat{S}_{D,-}\widehat{S}_{T,-}$, $\widehat{S}_{D,+}\widehat{S}_{T,z}$ and $\widehat{S}_{D,-}\widehat{S}_{T,z}$ of the dipolar coupling were also ignored. They connect spin states of different manifolds of the doublet spin and the corresponding energies cannot be significantly shifted by the comparably weak dipolar coupling. It is shown in S2 that at magnetic field strengths that are relevant for experimental conditions the included non-secular terms from Eq. (S3) are sufficient and no further distortions are to be expected by the left-out ones.

We also deleted these sentences from S1, p.3, l.4-7 to avoid repetition:

~~Here, $\widehat{S}_{T,+}$ and $\widehat{S}_{D,+}$ are the raising ($\widehat{S}_{+} = \widehat{S}_{x} + i\widehat{S}_{y}$) and $\widehat{S}_{D,-}$ and $\widehat{S}_{T,-}$ are the lowering ($\widehat{S}_{-} = \widehat{S}_{x} - i\widehat{S}_{y}$) operators. Note that this non-secular part does not include all non-secular terms and all-pseudo secular terms are also ignored here. As is shown in S2 these remaining terms have only a negligible effect at the magnetic fields strengths that are relevant in most experiments, and this is why they can be left out.~~

15. All of the simulations aimed at illustrating the effects of different spin system and experimental parameters on the dipolar trace have been performed for a single distance and show modulations that persist for a very long time. All of the systems encountered experimentally will be characterized by a distribution of distances. I believe it would be more instructive and more meaningful to show simulations performed for distance distributions in addition to or instead of the single-distance simulations as these would be closer to traces encountered experimentally and allow users to judge what effect they could expect in experimentally relevant situations. For example, it would be interesting to see how the V_mS=0 contribution would be affected by distance distributions with increasing widths. Also, would the effects illustrated in Figures 6, 8 and S8 even be visible in the presence of a distance distribution rather than a single distance?

We performed additional simulations with an entire distance distribution with different widths. We added the following paragraph starting from p.15 l.14 to:

As can be seen in Figure 1 the width of the distance distribution also has an influence on the decay of $V_0^{\text{non-sec}}(t)$. In X-band (Figure 1a) and for small standard deviations of $\sigma = 0.05$ nm, $V_0^{\text{non-sec}}(t)$ has a sigmoid like shape. Increasing the width has a twofold effect on the decay of $V_0^{\text{non-sec}}(t)$. Whereas the initial decay is steeper, on a long scale, the decay of $V_0^{\text{non-sec}}(t)$ is decreased for broader distance distributions. This can clearly be seen in the case of $\sigma = 3$ nm where for $t < 1$ µs $V_0^{\text{non-sec}}(t)$ decays faster for the simulation with $\sigma = 3$ nm than with $\sigma = 0.05$ nm, but for $t > 1$ µs $V_0^{\text{non-sec}}(t)$ decays slower for $\sigma = 3$ nm than for $\sigma = 0.05$ nm. In Q-band where the decay of $V_0^{\text{non-sec}}(t)$ is generally slower, the simulations in Figure 1b show that here only the first effect is of relevance. It can be seen that the first part of the decay of $V_0^{\text{non-sec}}(t)$ is again steeper for broader distance distributions, but the second part where this behavior is inverted lies outside the time window. This means that in Q-band the width of the distance distribution has a smaller influence on the decay of $V_0^{\text{non-sec}}(t)$ than in Q-band.

[Figure]

Figure 1: The influence of the width of the distance distribution on the decay of $V_0^{non-sec}(t)$ for TPP in a) X-band and b) Q-band. The simulations were performed for a Gaussian distance distribution width a mean of 3 nm and different standard deviations $\sigma$.

We do not think that the effect of the ZFS would be visible in Fig. 6, 8 and S8. We added the following sentences on p.17, l.6:
However, for experimentally relevant distance distributions with a finite width, the oscillations typically fade out much quicker and cases where four oscillations can be resolved are scarce. In such a case, the observed influence of the ZFS for high values of $q$ can be expected to almost negligible.

And we added the following sentence on p.21, l.23:
However, for experimentally relevant cases with distance distributions of a finite width, the oscillations in the dipolar trace fade out much faster anyway. It is to be expected that in these cases, the effect of the ZFS on the LiDEER trace are rather small and that therefore artifacts in the distance distribution are not so pronounced, even in the case when the observer pulses are set to a non-canonical orientation.

16. In the initial discussion of LiDEER, transition selection is mentioned, but orientation selection is currently not. In the discussion of the LiDEER simulations and orientation selection, it would be useful to more clearly separate discussion of the effect of orientation selection on its own and the orientation selection effect on the additionally considered non-secular contributions. The sentence in lines 14-15 of page 20 will likely not be clear to readers not familiar with orientation selection in triplet states, I would consider including a plot to illustrate what is meant.

[revised manuscript text omitted]

A new section S11 was added in the SI:

**S11 Analysis of the experimental LaserIMD data**

[Figure]

**Figure S6:** Experimental LaserIMD data of TPP-pAA$_5$-NO• recorded in X-band at 30 K in MeOD/D$_2$O (98/2 vol.%). a) Analyzed with a kernel that includes the ZFS and b) Analyzed with a kernel that ignores the ZFS. The raw data are depicted on the left side as grey dots with the fits as green line, the background fit is depicted as dashed grey line. The distance distributions obtained with Tikhonov regularization (Fábregas Ibáñez et al., 2020) is shown on the right side. The shaded areas correspond to the 95% confidence intervals that were obtained with bootstrapping.

[Figure]

**Figure S7:** Experimental LaserIMD data of TPP-pAA$_5$-NO• recorded in Q-band at 30 K in MeOD/D$_2$O (98/2 vol.%). a) Analyzed with a kernel that includes the ZFS and b) Analyzed with a kernel that ignores the ZFS. The raw data are depicted on the left side as grey dots with the fits as blue line, the background fit is depicted as dashed grey line. The distance distributions obtained with Tikhonov regularization (Fábregas Ibáñez et al., 2020) is shown on the right side. The shaded areas correspond to the 95% confidence intervals that were obtained with bootstrapping.

[Figure]

**Figure S8:** Experimental LaserIMD data of TPP-pAA$_{10}$-NO• recorded in X-band at 30 K in MeOD/D$_2$O (98/2 vol.%). a) Analyzed with a kernel that includes the ZFS and b) Analyzed with a kernel that ignores the ZFS. The raw data are depicted on the left side as grey dots with the fits as red line, the background fit is depicted as dashed grey line. The distance distributions obtained with Tikhonov regularization (Fábregas Ibáñez et al., 2020) is shown on the right side. The shaded areas correspond to the 95% confidence intervals that were obtained with bootstrapping.

[Figure]

**Figure S9:** Experimental LaserIMD data of TPP-pAA$_{10}$-NO• recorded in Q-band at 30 K in MeOD/D$_2$O (98/2 vol.%). a) Analyzed with a kernel that includes the ZFS and b) Analyzed with a kernel that ignores the ZFS. The raw data are depicted on the left side as grey dots with the fits as orange line, the background fit is depicted as dashed grey line. The distance distributions obtained with Tikhonov regularization **(Fábregas Ibáñez et al., 2020)** is shown on the right side. The shaded areas correspond to the 95% confidence intervals that were obtained with bootstrapping.

**Table S2: Background decay rates and modulation depths as obtained by the analysis of the LaserIMD data of TPP-pAA$_5$-NO•.**

|  | X-band | | Q-band | |
|---|---|---|---|---|
|  | w/ ZFS | w/o ZFS | w/ ZFS | w/o ZFS |
| **Background decay rate [μs$^{-1}$]** | 0.0 (0.0, 0.2) | 0.1 (0.0, 0.3) | 0.000 (0.003, 0.000) | 0.33 (0.27, 0.36) |
| **Modulation depth [%]** | 57 (52, 59) | 48 (43, 52) | 71 (70, 72) | 46 (45, 49) |

**Table S6: Background decay rates and modulation depths as obtained by the analysis of the LaserIMD data of TPP-pAA$_{10}$-NO•.**

|  | X-band | | Q-band | |
|---|---|---|---|---|
|  | w/ ZFS | w/o ZFS | w/ ZFS | w/o ZFS |
| **Background decay rate [μs$^{-1}$]** | 0.00 (0.00, 0.05) | 0.13 (0.01, 0.28) | 0.03 (0.01, 0.04) | 0.09 (0.07, 0.11) |
| **Modulation depth [%]** | 43 (40, 45) | 31 (25, 36) | 26 (25, 27) | 17 (16, 18) |

  In absence of orientation selection, the orientation of the dipolar vector and the transient triplet label are not correlated and the integration over the corresponding Euler angles can be done independently. This is often realized in practical applications where flexible linkers are used to attach labels to the studied molecule.

- The authors should consider including the simulation script for LiDEER used for the time-domain simulations with Spinach in the SI.

  The source code for the LiDEER simulations was made available online at https://github.com/andreas-scherer/LiDEER_simulations.git. Th link was also added to the manuscript.

- On page 13, line 14, the authors state that the frequency shift "seems to be averaged out after integration". Would it be possible to expand on this or explain why?

  Our hypothesis is that is caused by the angular dependency of the LaserIMD frequencies and the integrals in Eq. 25-27. The frequencies $\omega_{+1}^{\mathrm{non-sec}}$ and $\omega_{-1}^{\mathrm{non-sec}}$ are a sum of the secular part $3\cos\left(\beta_{\mathrm{dip}}\right)^2 - 1$ and the non-secular part $\delta_{\mathrm{ZFS}}\sin(2\beta_{\mathrm{d}})$ (Eq. 21 and Eq. 23). As in the powder average over the orientation of the dipolar coupling vector, the orientations are weighted with $\sin(\beta_{\mathrm{dip}})$, the perpendicular orientation of the dipolar vector with $(\beta_{\mathrm{dip}} = \pi/2)$ contributes the most. At this orientation, the secular term $3\cos\left(\beta_{\mathrm{dip}}\right)^2 - 1$ is equal to 2, whereas the non-secular term $\delta_{\mathrm{ZFS}}\sin\left(2\beta_{\mathrm{dip}}\right)$ is equal to 0. Therefore, at the angle $\beta_{\mathrm{dip}}$ that contributes most to the signal the non-secular terms -and with them the influence of the ZFS- vanishes. Furthermore, it must be considered that with the used parameters $|\delta_{\mathrm{ZFS}}| \approx 0.1$, which means that the non-secular part $\delta_{\mathrm{ZFS}}\sin\left(2\beta_{\mathrm{dip}}\right)$ is approximately one order of magnitude smaller than the secular part $3\cos\left(\beta_{\mathrm{dip}}\right)^2 - 1$. Therefore, the ZFS leads to a shift of the frequency $\omega_{+1}^{\mathrm{non-sec}}$ and $\omega_{-1}^{\mathrm{non-sec}}$ that is relatively small compared to the frequency $\omega_{+1}^{\mathrm{sec}}$ and $\omega_{-1}^{\mathrm{sec}}$ and $V_{+1}^{\mathrm{non-sec}}(t)$ and $V_{-1}^{\mathrm{non-sec}}(t)$ are not so much influence by the ZFS. In $V_0^{\mathrm{non-sec}}(t)$, on the other hand, the LaserIMD frequency $\omega_0^{\mathrm{non-sec}}$ consists only of the non-secular part (Eq. 21). As $\omega_0^{\mathrm{sec}} = 0$, the ZFS dependent non-secular part dominates the trace $V_0^{\mathrm{non-sec}}(t)$ and its influence is much stronger than for other two coherencetransfer pathways. Taken together, these arguments can explain why an effect of the ZFS is seen in $V_0^{non-sec}(t)$ but not in $V_{+1}^{non-sec}(t)$ and $V_{-1}^{non-sec}(t)$. However, a more rigorous mathematical explanation is missing and therefore we did not include that in the manuscript.

- On page 13, the authors state that modulation depths larger than 66.6% could be reached as the "modulation depth is increased by the ZFS". Given that the additional contribution from ms=0 appears to be a decay rather than a modulation, can it really be considered an increase in modulation depth?

For a single orientation the contribution from $m_s = 0$ is a sine/cosine oscillation in the complex plane just as it is the case for the other two contributions. The decay only emerges as a result of the powder average. In the case of DEER with broad distance distributions for example, the DEER trace typically also does not show clear modulations and consists only of a decaying signal. Therefore, we think that from a purely conceptual point it is still appropriate to consider the signal of $V_0^{non-sec}(t)$ as a contribution to the modulation depth in LaserIMD.

- The effects on the distance distributions in Figure 7 are hard to make out in the current plots.

We have changed this plot to:

[Figure]

- The authors consider two different sets of spin system parameters for their simulations, with a D parameter of ca. 1 GHz, which is assigned to TPP, and with a D parameter of 3.5 GHz, for which they only vaguely state that "such high values are possible for some labels". I would recommend specifying which label(s) they mean.

We changed this to:
Such high values are possible for some labels like Rose Bengal or Erythrosin B (Williams et al., 2020; Bertran et al., 2022b).

- When the software tool for the calculation of LaserIMD kernels is mentioned, it would be useful to also mention this can be used in conjunction with DEERLab.

We added the following sentence on p23 line 11:

The obtained kernel can for example be used in combination with the software DeerLab (Fábregas Ibáñez et al., 2020) to analyze experimental LaserIMD traces.

- In Figure S8, the authors should specify more clearly and indicate where the observer pulse was placed with respect to the triplet spectrum.

  The position of the observer pulse with respect to the triplet spectrum is shown in Fig. S7. A reference to this figure was added in the description of Fig. S8:
  The position of the observer and pump pulse with respect to the EPR spectrum is shown in Fig. S7a, c and e.

20. Typos:

We corrected all typos. In particular, we

- clarified p.3 line 1 be rephrasing the paragraph:
  In previous works, LaserIMD and LiDEER data were analyzed under the assumption that the zero-field splitting interaction (ZFS) can be ignored (Di Valentin et al., 2014; Hintze et al., 2016; Bieber et al., 2018; Dal Farra et al., 2019a; Krumkacheva et al., 2019).Under this assumption  the dipolar traces of LaserIMD and LiDEER have the same shape as  for DEER on a label pair with two $S = 1/2$ spins . However, as is shown below, this assumption is only correct if all spin-spin interactions are much smaller than the Zeeman-interaction with the external magnetic field.

- unified the usage of $\omega_\mathrm{D}$ and $\omega_\mathrm{T}$. Where they were used to describe frequencies we renamed them to $\nu_\mathrm{D}$ and $\nu_\mathrm{T}$. Where they were used to mean angular frequencies, we replaced them with $2\pi\nu_\mathrm{D}$ and $2\pi\nu_\mathrm{T}$.

- rephrased p.15, lines 8-9:
  As can be seen in Eq. (21) -(23), changing the distance $r$ from $2.2$ to $5\ \mathrm{nm}$ leads to an increase of the LaserIMD frequencies $\omega_{+1}^{\mathrm{non-sec}}$, $\omega_0^{\mathrm{non-sec}}$ and $\omega_{-1}^{\mathrm{non-sec}}$ that scales with $r^{-3}$.

- Changed the y axis of Figure 7 (now Figure 8).

- rephrased p.19, line 14:
  In Q-band the fitted background decay is always a bit larger than the true value. Except for the case were the true background decay is set to $k = 0\ \mathrm{\mu s^{-1}}$, the deviation of the fitted and the true background decay is smaller in Q-band than in X-band.

- rephrased p.21, lines 21-22:
  Here, the effect of the ZFS can clearly be seen and the LiDEER trace of the simulation with $D = 3500\ \mathrm{MHz}$ and $E = -800\ \mathrm{MHz}$ in X-band shows strong deviations from the other traces that were simulated with a smaller ZFS.

- rephrased page 5 of the SI after eq. S31:
  Insertion of Eq. (S16) an (S17) for $E_{\mathrm{ZFS}}^{\mathrm{non-sec}}$ and $E_{\mathrm{dip}}^{\mathrm{non-sec}}$ gives the expressions of Eq. (21) –(23) of the main text.